# DNA binding redistributes activation domain ensemble and accessibility in pioneer factor Sox2

Sveinn Bjarnason [1,5], Jordan A. P. McIvor[2,5], Andreas Prestel [3], Kinga S. Demény [1], Jakob T. Bullerjahn [4], Birthe B. Kragelund [3], Davide Mercadante [2] ✉ & Pétur O. Heidarsson [1] ✉

More than 1600 human transcription factors orchestrate the transcriptional machinery to control gene expression and cell fate. Their function is conveyed through intrinsically disordered regions (IDRs) containing activation or repression domains but lacking quantitative structural ensemble models prevents their mechanistic decoding. Here we integrate single-molecule FRET and NMR spectroscopy with molecular simulations showing that DNA binding can lead to complex changes in the IDR ensemble and accessibility. The C-terminal IDR of pioneer factor Sox2 is highly disordered but its conformational dynamics are guided by weak and dynamic charge interactions with the folded DNA binding domain. Both DNA and nucleosome binding induce major rearrangements in the IDR ensemble without affecting DNA binding affinity. Remarkably, interdomain interactions are redistributed in complex with DNA leading to variable exposure of two activation domains critical for transcription. Charged intramolecular interactions allowing for dynamic redistributions may be common in transcription factors and necessary for sensitive tuning of structural ensembles.

Transcription factors (TFs) consolidate information for gene expression by locating specific DNA sequences in the nucleus and recruiting cofactors to regulate transcription. Most human TFs consist of structured DNA binding domains (DBDs) and long intrinsically disordered regions (IDRs) that can harbour activation domains (ADs), and thus interaction sites for regulatory binding partners[1,2]. Whereas intense focus has been on the structured DBDs, IDRs in TFs have been understudied due to the major challenges such regions pose for traditional structural biology techniques. Consequently, there is a significant lack of accurate descriptions of IDR ensembles for all of roughly 1600 human TFs, both off- and on their DNA recognition sites. Beyond hosting the ADs important for transcriptional activation, IDRs

in TFs can have many other roles such as modulating DNA binding affinity[3], contributing competence for phase separation[4], or regulating DNA binding specificity[5]. In recent years, the importance of electrostatic interactions for the conformational dynamics of IDRs has become increasingly evident[6–8]. Experiments and computational modelling have suggested that charged patches on folded domains modulate the dimensions of adjacent IDRs, which might have direct functional consequences[9–13], and charge modulation by posttranslational modifications (PTMs) such as phosphorylation can have a large impact on the ensemble[2,14]. However, the conformational signatures of such molecular behaviour have not been broadly established, and generally, IDR conformational dynamics and their modulation by DNA

[1]Department of Biochemistry, Science Institute, University of Iceland, Sturlugata 7, 102 Reykjavík, Iceland. [2]School of Chemical Science, University of Auckland, Auckland, New Zealand. [3]Department of Biology, REPIN and Structural Biology and NMR Laboratory, University of Copenhagen, Ole Maaløes Vej 5, 2200 Copenhagen, Denmark. [4]Department of Theoretical Biophysics, Max Planck Institute of Biophysics, Max-von-Laue-Straße 3, 60438 Frankfurt am Main, Germany. [5]These authors contributed equally: Sveinn Bjarnason, Jordan A. P. McIvor. ✉e-mail: davide.mercadante@auckland.ac.nz; pheidarsson@hi.is

binding is poorly understood. Structural models of IDR ensembles are critical to understand the code of transcriptional regulation and to decode how PTMs affect gene regulatory networks.

We addressed these challenges by studying the structure and dynamics of pluripotency factor Sox2, a prototypical TF, which plays a pivotal role in maintaining embryonic and neuronal stem cells[15]. Sox2 is classified as a pioneer TF due to its ability to target its cognate binding sequence in condensed, nucleosome-rich DNA[16]. Sox2´s pioneer activity– along with the other so-called Yamanaka TFs Oct4, Klf4, and c-Myc–, has recently been applied to generate induced pluripotent stem cells (iPSCs), bringing immense potential to regenerative medicine and drug development[17]. Sox2 has 317 residues and consists of a

small HMG-box DBD[18] flanked N-terminally by a short 40-residue low-complexity stretch and C-terminally by a long ~200-residue region, both of which are predicted to be disordered (N-IDR and C-IDR, respectively) (Fig. 1a). Little is known about the function of the short N-IDR but there is evidence that it is important for interactions with other TFs[19]. The DBD is rich in positively charged residues (net charge = +13)– as commonly observed in DNA-binding proteins[2]– which facilitate binding to the negatively charged DNA. The C-IDR is enriched in methionines, serines, glycines and prolines (~40% of total residues) and contains 18 charged residues (zero net charge) distributed throughout the sequence. The C-IDR contains two predicted ADs: AD1 (residues ~150–200), which was recently validated in a large-

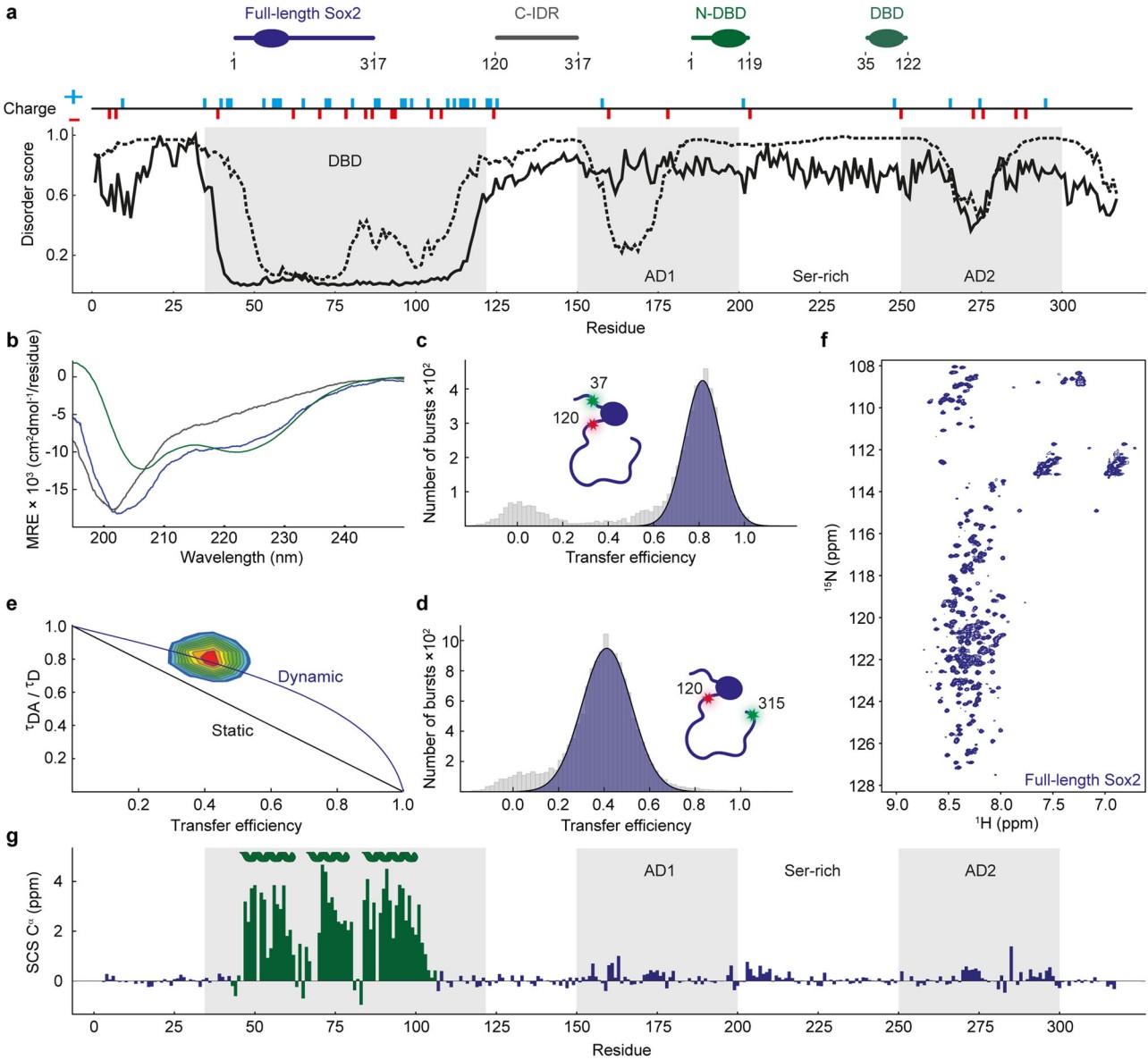

**Fig. 1 | Sox2 C-IDR is disordered and dynamic. a** Schematics of Sox2 illustrating the main constructs used in this study. The plot shows disorder predictions as a function of residue number, based on two different predictors (Disopred3[77] (dashed line), AlphaFold[22] normalised pLDDT (solid line)). The DBD is indicated, as are the ADs and serine-rich region (see text for details), and the locations of charged residues. **b** Far-UV circular dichroism spectra of different Sox2 variants at 5 µM concentration; Full-length Sox2 (blue), C-IDR (grey), N-DBD (green). Spectra are averages of $n = 3$ independent measurements. **c**, **d** Single-molecule transfer efficiency histograms of Sox2 fluorescently labelled flanking the DBD (residues 37 and 120, number of molecules=5323) or probing the entire C-IDR (residues 120–315,

number of molecules = 14,544). The small peak at E - 0 originates from donor-only labelled molecules that remain after filtering (see Methods and Supplementary Fig. 1). **e** Fluorescence lifetime analysis of the Sox2 C-IDR. The 2D-correlation plot shows fluorescence lifetimes of the Cy3b donor ($\tau_{DA}$) relative to the intrinsic donor fluorescence ($\tau_D$). The dynamic line is based on a SAW-$v$ polymer model. See text for details. **f** $^1$H$^{15}$N-HSQC spectrum of full-length Sox2. **g** C$^\alpha$ SCS plot of full-length Sox2 (blue). SCSs for the DBD (green) were determined for the isolated N-DBD domain. The known helix locations (UniProt P48431) are indicated, and grey shaded areas indicate the DBD and ADs. Source data are provided as a Source Data file.

scale mapping of TF IDRs[20], and AD2 (residues ~250–300)[21–24] (Fig. 1a). The two ADs are separated by a serine-rich domain (residues ~200–250), which mediates direct interaction with the TF Nanog in a process important for self-renewal of embryonic stem-cells[25]. There is evidence that the IDRs of Sox2 are neccessary for pioneering function[26] but it is unclear whether they are important only for transcriptional activation or for other functions such as chromatin binding or opening, as observed for some pioneer factors[27,28]. Indeed, the C-IDR of Sox2 has recently been found to have functions that extend beyond transcriptional activation, ranging from contributing to force exertion on DNA[29], RNA binding[26,30], and DNA scanning and target site selection[31]. However, a quantitative description of the C-IDR conformational ensemble is lacking and it is unclear how the ensemble is affected by DNA binding, and ultimately how it conveys function.

In this work we used single-molecule Förster resonance energy transfer (smFRET) and nuclear magnetic resonance (NMR) spectroscopy, combined with molecular dynamics (MD) simulations to comprehensively map the conformational dynamics of full-length Sox2. We show that the C-IDR engages in dynamic interactions with the DBD involving its charged residues and that this constrains its dimensions in an exquisitely salt-sensitive manner. These interactions are substantially altered in complex with both DNA and nucleosomes which leads to a more extended C-IDR. We reconstruct experimentally-derived FRET values from a coarse-grained (CG) simulation and reveal the structural ensemble of free and DNA-bound Sox2. Our structural ensemble reveals a large-scale re-arrangement in the C-IDR dimensions upon DNA binding, which specifically redistributes the accessibility of the two transcriptional ADs. Considering general sequence features of TFs[2], this type of charge-driven IDR ensemble modulation is likely to be common among eukaryotic TFs where charge patterning and PTMs are expected to play an important role.

## Results

### Sox2 C-IDR is disordered and dynamic

While structures of the Sox2 DBD show that its conformations in free and DNA-bound states are highly similar[32,33], high-resolution structural information on full-length Sox2 in regions outside the DBD are currently unavailable. Structure and disorder predictions indicate that the mainly disordered C-IDR contains short polypeptide stretches with some secondary structure propensities which coincide with the ADs (Fig. 1a). Indeed, far-UV circular dichroism (CD) spectra of full-length Sox2 as well as of isolated domains (N-terminal domain and DBD (N-DBD), and C-IDR) generally agree with predictions (Fig. 1b, Supplementary Table 1). The far-UV CD spectrum of the N-DBD showed minima at 222 nm and 208 nm, suggesting the presence of mainly helices, whereas the C-IDR gave a spectrum that suggested mainly a random-coil with a large negative ellipticity minimum at 202 nm, indicating an overall lack of secondary structure.

To quantify the dimensions and dynamics of Sox2 in more detail we turned to smFRET[34,35]. We designed cysteine mutations to specifically probe the major domains and labelled them through thiol chemistry using the fluorophore pair Cy3b and CF660R. We then used smFRET to measure mean transfer efficiency, $\langle E \rangle$, of thousands of individual and freely-diffusing molecules using a confocal fluorescence microscope. When the dyes were flanking the DBD (positions 37 and 120, Fig. 1c) we measured an $\langle E \rangle$ ~ 0.8, which corresponds to an average distance between the dyes close to that expected from structural studies (PDB 6T7B), indicating that the DBD remains folded in our experiments (Methods and Supplementary Table 2). For probing the long C-IDR, we placed the dyes just after the DBD (position 120) and near the C-terminus (position 315), measuring a FRET efficiency $\langle E \rangle = 0.43$ (Fig. 1d). Given that the structure predictions and CD data indicate a mainly random coil for the C-IDR, we used a self-avoiding walk polymer model with a variable scaling exponent $v$ (SAW-$v$) to determine the root mean square distance ($R_{RMS}$) between the two dyes

(Methods). The SAW-$v$ model has recently been shown to describe well the dimensions of intrinsically disordered proteins (IDPs)[36]. The $\langle E \rangle$ of the C-IDR leads to an $R_{RMS}$ of 7.5 nm and a scaling exponent $v$ of 0.57, which is within the range expected for an IDP[37].

To probe rapid conformational dynamics of the C-IDR, we can use relative fluorescence lifetimes to detect distance fluctuations between the two fluorophores, on a timescale between the fluorescence lifetime (ns) and the interphoton time (µs). The relative donor lifetime (the ratio between the donor lifetime in absence ($\tau_D$) and presence ($\tau_{DA}$) of an acceptor) can be shown from the Förster equation to equal to $\frac{\tau_{DA}}{\tau_D} = 1 - \langle E \rangle$ only if there is a single, effectively static distance (on the same timescale) separating the two dyes (Fig. 1e and Methods). Conversely, if a distribution of distances is sampled due to dynamics of the polypeptide chain, the relative lifetimes cluster above the diagonal line, to an extent defined by the variance of the underlying distance distribution. For dyes probing the Sox2 C-IDR, the relative lifetimes deviate significantly from the diagonal "static" line and agree with a "dynamic" line based on the expected behaviour of a SAW polymer with a scaling exponent of 0.57, as obtained from the measured $\langle E \rangle$.

Since the FRET experiments do not report directly on potential secondary structure formation, we used NMR spectroscopy to extract residue-specific structural information on Sox2. We produced $^{15}N^{13}C$-isotope labelled full-length Sox2 and first measured a $^1H^{15}N$-heteronuclear single quantum coherence (HSQC) spectrum of full-length Sox2. The HSQC spectrum displayed almost the full set of expected signals from all backbone amides (Fig. 1f), with little dispersion of resonances in the proton dimension, characteristic of an IDR[6]. From sets of triple resonance spectra, we could assign 275 peaks out of 290 assignable (>95%). The peak intensities of residues in the DBD were much lower than for the disordered regions, presumably due to slow rotational tumbling, hence the assignments of the DBD NMR signals were performed for the isolated N-DBD and transferred to the spectra of full-length Sox2 (Fig. 2, Supplementary Fig. 2, and Methods). A secondary chemical shift (SCS) analysis of $C^\alpha$ and $C^\beta$ shifts revealed a general lack of secondary structures in the C-IDR with potential transient helix or turn formation in regions coinciding with the ADs (<7% helix in residue regions G150-Q175, Y200-S220, S275-S300, calculated using the shifts for the DBD as reference for 100%) in agreement with predictions, whereas we observed strong signatures for the three expected helices in the DBD (Fig. 1a, g).

### C-IDR dimensions are shaped by charged interactions with the DBD

The classical modular view of TFs, which assumes separate functional domains unaffected by each others' presence, has recently come into question and at the same time, interdomain synergy and context are increasingly coming into view[38,39]. Charged residues can partake in long-range interactions and play a primary role in the conformational dynamics of IDRs[6]. The fraction of charged residues in the C-IDR of Sox2 (+9,-9) classifies it as a weak polyampholyte and predicts it to adopt a collapsed state[40]. However, the DBD contains a high density of charges, with a net charge of +13 to facilitate binding with the negatively charged DNA. We therefore investigated whether interactions between the C-IDR and the neighbouring DBD might contribute to the observed dimensions of the C-IDR. We produced fluorescently labelled isolated DBD and C-IDR to compare their dimensions to that of the full-length protein using smFRET. We used a Sox2 construct with fluorophores in positions 120 and 265, which probes the majority of the C-IDR with high sensitivity ($\langle E \rangle = 0.55$, which is close to the Förster radius at $E = 0.5$). We observed a significantly lower FRET efficiency for the isolated C-IDR compared to the same region within full-length Sox2 ($\langle E \rangle = 0.48 \pm 0.01$ vs. $\langle E \rangle = 0.55 \pm 0.01$, respectively) (Fig. 2a), whereas the end-to-end distance of the DBD (fluorescently labelled in residues 37 and 120) was largely independent of context (Fig. 2b). These data indicate that the C-IDR is more compact in the presence of the

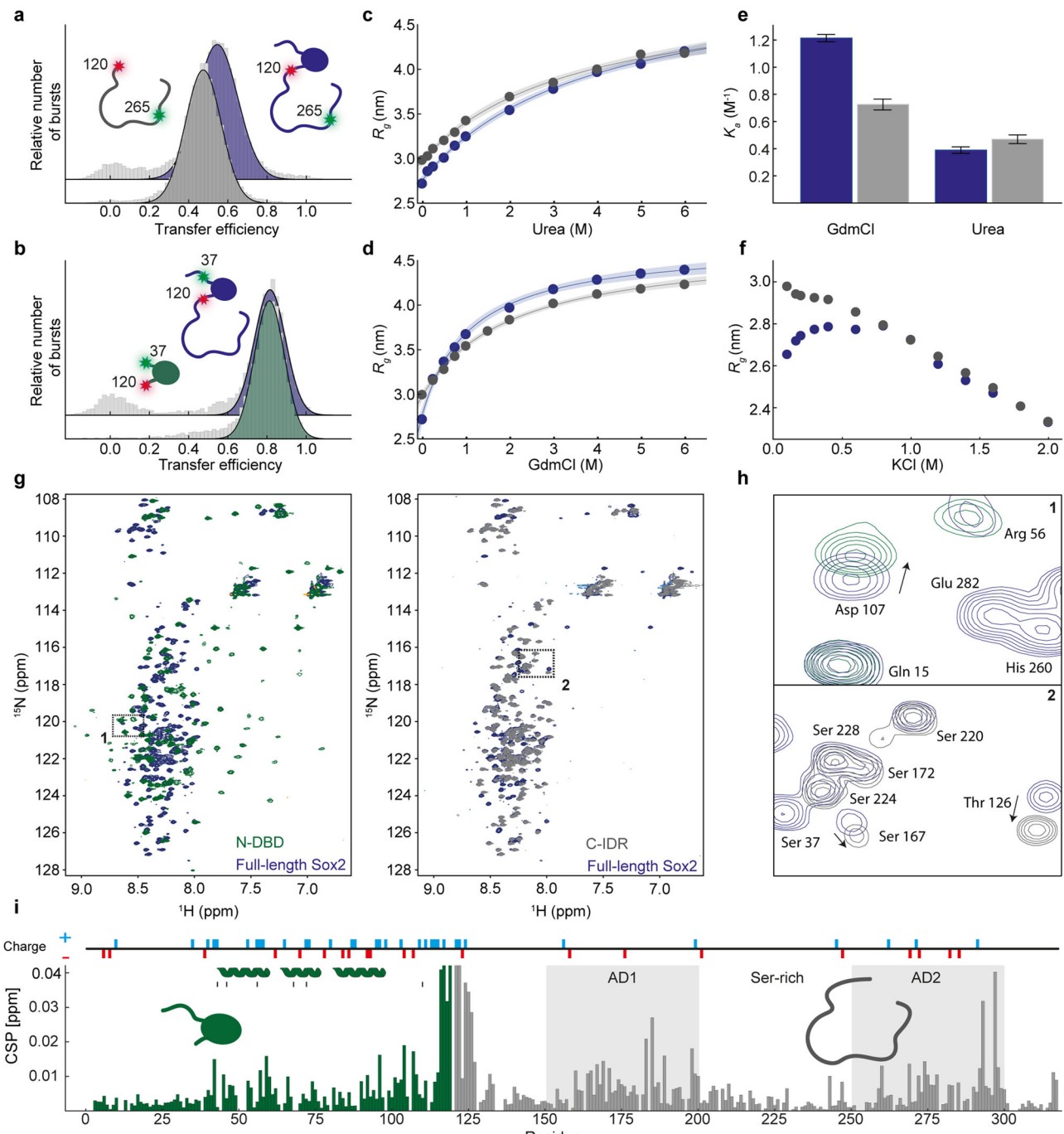

**Fig. 2 | Interdomain interactions between Sox2 DBD and C-IDR.** Single-molecule transfer efficiency histograms of full-length Sox2 and either an isolated C-IDR, both labelled at positions 120 and 265 (**a**), or an isolated DBD, both labelled at positions 37 and 120 (**b**). Apparent $R_g$ (see text and Methods for details) of the fluorescently labelled C-IDR in full-length Sox2 (blue) or isolated (grey) as a function of urea (**c**) or GdmCl (**d**) concentration. Each data point is derived from the mean of >5000 individual molecules. The solid lines are fits to a weak denaturant binding model and the shaded areas represent 95% confidence intervals centred on the fit line. **e** Denaturant association constant ($K_a$), determined from fits to the data in (**c, d**) for the C-IDR in full-length Sox2 (blue) and isolated (grey). Error bars are standard

errors of the fits in (**c, d**). **f** Apparent $R_g$ of the C-IDR in full-length Sox2 (blue) or isolated (grey) as a function of KCl concentration. **g** $^1H^{15}N$ HSQC spectra of full-length Sox2 (blue), overlayed with a spectrum of the isolated N-DBD (green, left) and the isolated C-IDR (grey, right). **h** Boxes 1 and 2 are zooms into specific regions of the HSQCs in (**g**), showing overlap of some peaks and changes in position of others. **i** CSP plot showing the chemical shift difference between full-length Sox2 and each isolated domain, N-DBD (green) and C-IDR (grey). Helix locations in the DBD are indicated where black lines denote residues important for DNA binding (PDB 6T7B). Grey shaded areas indicate ADs. Source data are provided as a Source Data file.

neighbouring N-DBD, providing strong evidence for the presence of interdomain interactions between the DBD and C-IDR.

To capture the physical basis for the interactions, we performed titration experiments by measuring FRET histograms in varying concentrations of chemical denaturants (urea or guanidinium

chloride (GdmCl)) or salt (KCl). The apparent radius of gyration, $R_g$ (determined from the SAW-$\nu$ distance distribution using the measured $\langle E \rangle$ at each denaturant concentration), was plotted as a function of titrant concentration (Fig. 2c, d, Supplementary Fig. 3). In both urea and GdmCl, the C-IDR gradually expanded (increased $R_g$)

with increasing concentration of denaturant for both the full-length protein and the isolated domain. We fitted the unfolding data with a weak denaturant binding model that assumes $n$-independent binding sites for denaturant molecules, which allows determination of an effective association constant, $K_a$ (see Methods). Interestingly, while the $K_a$ for urea, which is uncharged, is unaffected by the absence of the neighbouring N-DBD, the $K_a$ for GdmCl, which is charged, is reduced by almost 50% (Fig. 2e). Since the charged GdmCl disrupts electrostatic interactions whereas urea does not, this suggests the presence of interdomain communication between the DBD and C-IDR being based predominantly on interactions between charged residues. This was further supported when we measured transfer efficiency histograms over a range of salt concentrations (Fig. 2f, Supplementary Fig. 3). Remarkably, the C-IDR dimensions in full length Sox2 were exquisitely sensitive in the physiologically relevant range of salt concentrations (100–200 mM KCl). The full-length Sox2 displayed a pronounced "roll-over", suggesting screening of charge interactions with increasing salt concentrations, but the roll-over effect was entirely absent in the isolated C-IDR. Similar observations have been reported in other proteins[41] and can be explained by polyampholyte theory[42,43]; strong interactions between oppositely charged residues cause a collapse of the chain which are subsequently screened upon addition of salt, causing the chain to expand. The chain then compacts again at higher and unphysiological salt concentrations (700–2000 mM), potentially due to an enhancement of hydrophobic interactions as observed for other charged proteins[41]. Overall, even though the C-IDR contains relatively few charges causing it to adopt a collapsed state[40], charged interactions with the DBD sensitively control its dimensions further.

Long-range interdomain contacts should be revealed by differences in NMR chemical shifts between the full-length protein and isolated domains. We therefore produced $^{15}$N,$^{13}$C-isotope labelled isolated N-DBD and C-IDR for chemical shift assignments using sets of triple resonance NMR spectra. For the N-DBD and C-IDR we could assign 104 peaks out of 109 (expected excluding prolines and N-terminal methionine, >95%) and all 180 observable peaks in the $^1$H$^{15}$N-HSQCs, respectively (Fig. 2g, h). Comparing SCSs between the isolated C-IDR and the full-length protein revealed similarly lacking propensity to form secondary structure outside the DBD (Supplementary Fig. 4). The spectrum of the DBD displayed dispersed peaks, indicating a well-folded domain. Importantly, the C-IDR peaks overlapped well with the peaks from the full-length Sox2 in some regions but not in others, indicating a different chemical environment due to missing interdomain interactions in the isolated constructs, in agreement with the smFRET data (Fig. 2a). The regions with the largest chemical shift perturbations (CSPs) overlapped with regions of the highest charge density (Fig. 2i), in the vicinity of the ADs. The N-DBD was similarly affected mostly in the folded HMG domain that contains the highest density of charge, and in the region in close proximity to the missing C-IDR, whereas the N-terminal tail was minimally perturbed. These results were re-enforced by titrating a $^{15}$N-labelled C-IDR with an unlabelled DBD and vice versa, which showed considerable CSPs around the most charge-dense regions in both domains (Supplementary Fig. 4). Using the chemical shift changes of highly perturbed residues, we could estimate the dissociation constant, $K_D$, for the complex in trans to be $80 \pm 4\,\mu$M (Supplementary Fig. 4).

### DNA and nucleosome binding expands dimensions of C-IDR
Having established the conformational dynamics and interdomain interactions in the free state of Sox2, we next asked how these might be affected by complex formation with DNA. We speculated that perturbation of electrostatic interactions across domains upon DNA binding would lead to conformational changes in the C-IDR. We first checked

that Sox2 binding leads to the expected bending of DNA[32] by using fluorescently labelled oligonucleotides carrying a Sox2 binding site (TTGT) (Supplementary Table 3). At physiological salt concentrations (165 mM KCl), the free 15 bp dsDNA had a FRET efficiency $\langle E \rangle$ ~0.4 (Fig. 3a). When unlabelled Sox2 was added to the solution, another population appeared at higher FRET, $\langle E \rangle$ ~0.6, indicative of the expected Sox2-mediated DNA bending. We used the areas of the resulting FRET histograms to determine the fraction of bound DNA as a function of Sox2 concentration, and thus estimated the equilibrium dissociation constant, $K_D$. We constructed and fitted binding isotherms for both full-length Sox2 and the isolated DBD, and observed that the dissociation constant was largely unaffected by the presence of the C-IDR ($0.3 \pm 0.1$ nM for DBD vs $0.4 \pm 0.2$ nM for full-length Sox2), in agreement with previous results[26,30] (Fig. 3b, Supplementary Fig. 5, and Supplementary Table 4). This was also true for a non-specific DNA without a Sox2 binding site yet with ~10-fold higher $K_D$, also in agreement with previous results (Supplementary Fig. 5). Thus, both specific and non-specific DNA binding to the DBD was unaffected by the interdomain interaction. The dissociation constant determined using fluorescently labelled Sox2 (Supplementary Fig. 5) was very similar to that obtained with labelled DNA, excluding adverse effects on binding affinity due to the fluorophores.

To detect potential changes to the C-IDR conformations when in complex with DNA, we measured single-molecule transfer efficiency histograms for Sox2 fluorescently labelled in the C-IDR and in presence of unlabelled target DNA (Fig. 3c). We observed a substantial change in FRET efficiency; the C-IDR expanded considerably upon binding DNA, with FRET decreasing from 0.43 to 0.28 (Fig. 3c). This is in contrast to the DBD end-to-end distance which even compacted slightly (Supplementary Fig. 5). The change in FRET corresponds to an increased $R_{RMS}$ for the C-IDR ensemble from 7.5 nm to 9.2 nm or more than 20%. Analysis of the relative lifetimes of fluorophores probing the C-IDR in complex with DNA still showed deviation from a static distance, indicating that submillisecond dynamics of the C-IDR persist on DNA (Fig. 3e). To quantify the dynamics, we performed nanosecond fluorescence correlation spectroscopy (nsFCS) experiments of Sox2 in absence and presence of DNA, probing the C-IDR dynamics (Supplementary Fig. 6). Fitting the anti-correlated donor-acceptor cross-correlation functions, which decay on the timescale of interdye distance fluctuations, allowed us to determine the reconfiguration time ($\tau_r$) of the C-IDR (Methods). In agreement with the fluorescence lifetime analysis, $\tau_r$ is similar in the absence and presence of DNA (172 ns and 184 ns, respectively) whereas the isolated C-IDR reconfigures slightly faster ($\tau_r$ ~ 105 ns), presumably due to the lack of the neighbouring DBD to interact with.

Sox2 is a strong nucleosome binder, which is thought to play a role in its function as a pioneer factor. We therefore also tested whether similar conformational changes as observed for DNA would occur upon binding to nucleosomes. We reconstituted nucleosomes using the strongly positioning Widom-601 sequence with an incorporated Sox2 binding site, previously shown to be stably bound by Sox2[32] (Fig. 3d, Supplementary Fig. 7, Supplementary Table 3). We then measured transfer efficiency histograms for full-length Sox2 fluorescently labelled in the C-IDR and in the presence of unlabelled nucleosomes. The mean FRET efficiency of the C-IDR in complex with nucleosomes was very similar to the one measured in complex with a shorter DNA (Fig. 3c, d), and fluorescence lifetime analysis showed slightly dampened dynamics (Fig. 3f), which could indicate a weak interaction with the histone octamer. We confirmed that the DNA stays wrapped around the histone octamer during the experiment by estimating the diffusion time of Sox2 in the presence of DNA and nucleosomes, and by measuring FRET on fluorescently-labelled nucleosomes (Supplementary Fig. 7)[44]. Overall, these data thus indicate that the conformational ensemble of the Sox2 C-IDR is similar in complex with DNA and nucleosomes.

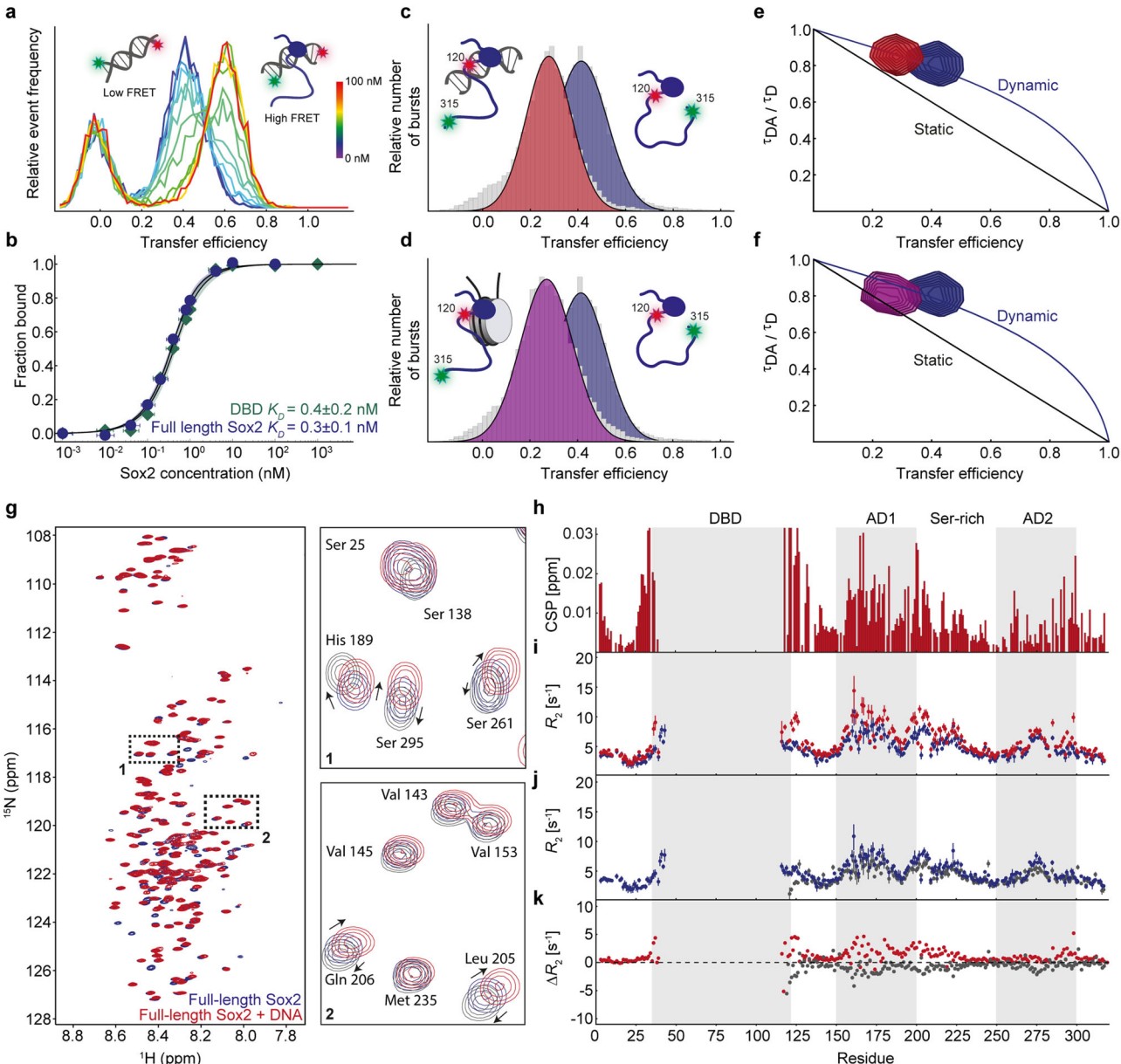

**Fig. 3 | Conformational rearrangements of the Sox2 C-IDR upon binding DNA and nucleosomes. a** Single-molecule transfer efficiency histograms of fluorescently labelled 15 bp DNA containing Sox2 binding site, with different concentrations of unlabelled full-length Sox2. The peak at E ≈ 0 corresponds to a population of molecules without an active acceptor. **b** The corresponding binding isotherms with fits (solid lines) to a 1:1 binding model, for both full-length Sox2 and the isolated DBD. Data are presented as mean values +/- SEM, estimated from dilution errors. Single-molecule transfer efficiency histograms of full-length Sox2 fluorescently labelled in the C-IDR, in the absence (blue) and presence of (**c**) 15 bp DNA (red) or (**d**) 197 bp nucleosomes (purple). Fluorescence lifetime analysis of Sox2 in the absence (blue) and presence of (**e**) DNA (red) or (**f**) nucleosomes (purple). **g** $^1$H$^{15}$N HSQCs of free Sox2 (blue) and Sox2 in complex with 15 bp unlabelled DNA (red). Zoomed-in regions show resonances that are affected or unaffected by DNA binding. Plots of (**h**) CSPs for Sox2 upon DNA binding and $^{15}$N-relaxation data (**i**–**k**) $R_2$ for free Sox2 (blue) and DNA-bound Sox2 (red, **i**), isolated C-IDR (grey, **j**), and the respective difference plot (C-IDR - free Sox2 (grey), DNA bound Sox2 - free Sox2 (red), **k**). Error bars indicate 95% confidence intervals centred on values obtained from the fitting procedure. Source data are provided as a Source Data file.

To probe DNA binding on a residue-specific level, we again used NMR spectroscopy. A $^1$H$^{15}$N-HSQC of DNA-bound full-length Sox2 showed similar low dispersion of peaks from the C-IDR but distinct chemical shift changes when compared with free Sox2 (Fig. 3g), whereas peaks from the DBD were entirely absent. When we plotted the CSPs as a function of residue sequence, we observed that most of the CSPs localise to the regions we had previously observed to make contacts with the DBD (Figs. 3h, 2i). Importantly, many of the chemical shifts imply a different structural ensemble for the C-IDR in the DNA bound state than for the free C-IDR construct (Fig. 3g, h), suggesting that it is not just a simple release of interactions with the DBD but

rather a different ensemble that is populated on DNA (Fig. 3g, *zooms*). We then measured the fast time scale dynamics of the different states using NMR. Residue-specific relaxation rates (Fig. 3i, j, k, Supplementary Fig. 8), which probe ps-ns dynamics, were generally low and globally increased slightly across the entire polypeptide chain upon DNA binding, indicating contributions due to slowed tumbling. Comparing relaxation rates between free full-length Sox2 and either DNA-bound or the isolated C-IDR showed little changes in dynamics on this timescale. Overall, the NMR data indicate that the C-IDR structural ensemble is different in complex with DNA yet it remains dynamic, in agreement with the fluorescence lifetime analysis.

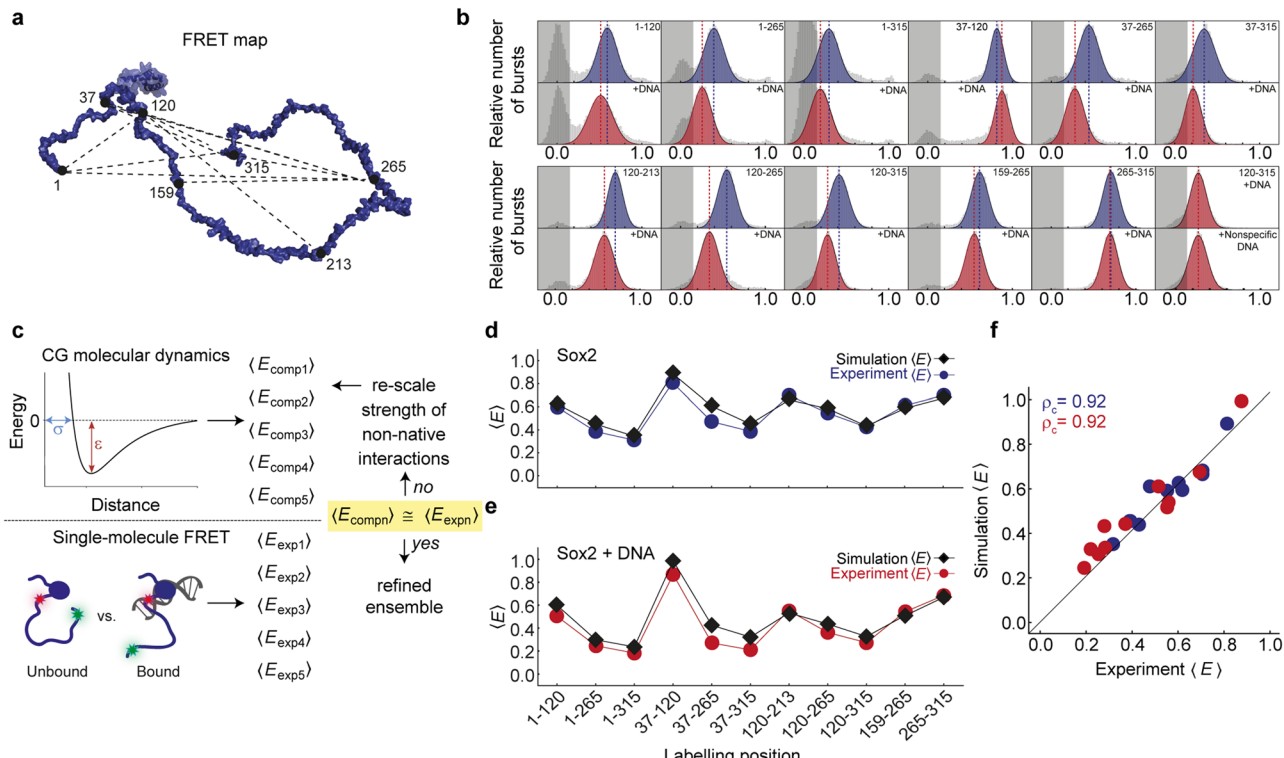

**Fig. 4 | Langevin dynamics simulations reproduce FRET efficiencies from smFRET experiments. a** Model of Sox2 showing FRET labelling positions that probe 11 unique intramolecular distances. **b** Single-molecule transfer efficiency histograms of free and DNA-bound Sox2 variants, fluorescently labelled in different positions. The last panels (bottom, right) show that transfer efficiency changes for fluorescently labelled C-IDR are identical with specific and non-specific DNA. **c** Schematic illustrating the CG computational approach. Using the Förster equation and a suitable polymer model, a series of computed FRET efficiencies (<*E*>) is obtained for each position labelled along the protein. The agreement between experimental and computed <*E*> is then refined by rescaling a single parameter ($\varepsilon_{pp}$) which uniformly defines the interaction strength between all beads in the intrinsically disordered domains, to finally obtain a refined ensemble. Comparison between computed (black) and experimentally-derived FRET efficiencies for (**d**) free (blue) and (**e**) DNA-bound (red) Sox2. **f** Correlations between experimentally-derived and computed FRET efficiencies for both free (blue) and DNA-bound (red) Sox2. High correlation coefficients are obtained for both free Sox2 and Sox2 bound to DNA ($\rho_c$ = 0.92). Solid line is the identity line. Source data are provided as a Source Data file.

## Coarse-grained simulation reveals redistributed accessibility of activation domains

To reconstruct the structural ensemble of Sox2 when free and bound to DNA, we performed CG Langevin dynamics simulations. Here, every amino acid is represented by a bead mapped on the $C^{\alpha}$ atom, while the DNA is represented by three beads resembling the ribose, base, and phosphate moieties. We used an integrative approach by which simulations aim to reproduce a series of experimentally obtained FRET efficiencies (*Methods*). For this purpose, we produced a set of additional fluorescently labelled Sox2 variants, designed to comprehensively probe discrete regions of the polypeptide chain, and measured transfer efficiency histograms and fluorescence lifetimes in the absence and presence of DNA (Fig. 4a, b, Supplementary Fig. 9). This yielded a total of 22 unique intramolecular FRET efficiencies (11 for free Sox2, 11 for bound Sox2), which were then matched in the simulations by tuning a single parameter, $\varepsilon_{pp}$, as it defines the interaction strength between the beads modelling the disordered regions of the protein (Fig. 4c). It is important to note that the simulation was performed at equilibrium, i.e., it was not restrained by the measured FRET efficiencies. Instead, FRET efficiencies were back-calculated from simulated distance distributions, and compared with the experiment afterwards. As in previous studies, the scalable interaction strength between beads was set to 0.4 kJ mol$^{-1}$ (0.16 kT) and gave the best match to the experimentally-derived FRET efficiencies (Fig. 4d, e). This approach has previously been shown to describe well the behaviour of several disordered proteins and protein–protein complexes with and without DNA[6,41,44–46]. Since no published structures are available for

free Sox2, we used the same structure for both free and DNA-bound Sox2[33]. However, a simulation using a thus far unpublished NMR structure of free Sox2 DBD deposited in the PDB (PDB code 2LE4) yielded near identical results (Supplementary Fig. 10).

The ensemble of both free and DNA-bound Sox2 collected from the simulated trajectories showed excellent agreement with the FRET efficiencies from experiments, yielding a concordance correlation coefficient $\rho_c$ of 0.92 (Fig. 4f). Given that interactions between beads within intrinsically disordered stretches are set to a minimal value, our simple CG model implies that a considerable driving force for contact formation between the IDRs and DBD comes from charged residues, in agreement with the FRET and NMR data (Figs. 2, 3). We thus investigated how salt affects the dimensions of Sox2 by simulating Sox2 in its free and bound states at apparent salt concentrations ranging from 20 to 400 mM (Supplementary Fig. 10). The ensemble of free Sox2 expands as a function of salt concentration (Supplementary Fig. 10) due to charge screening, but its dimensions reach a plateau at salt concentrations in proximity of the physiological range, in line with the experiments (Fig. 2f). Charge screening thus has an important effect on the dimensions of Sox2.

An analysis of the collected ensembles revealed a highly dynamic C-IDR that explores a range of different conformations but to different degrees depending on whether Sox2 is in its free or DNA-bound state (Fig. 5a, b, Supplementary Movies 1 and 2). In agreement with the experiments, the C-IDR dimensions are modulated by dynamic interactions with the DBD. Interestingly, an increase in salt concentration screens the interactions between the DBD and the AD1/AD2 domains

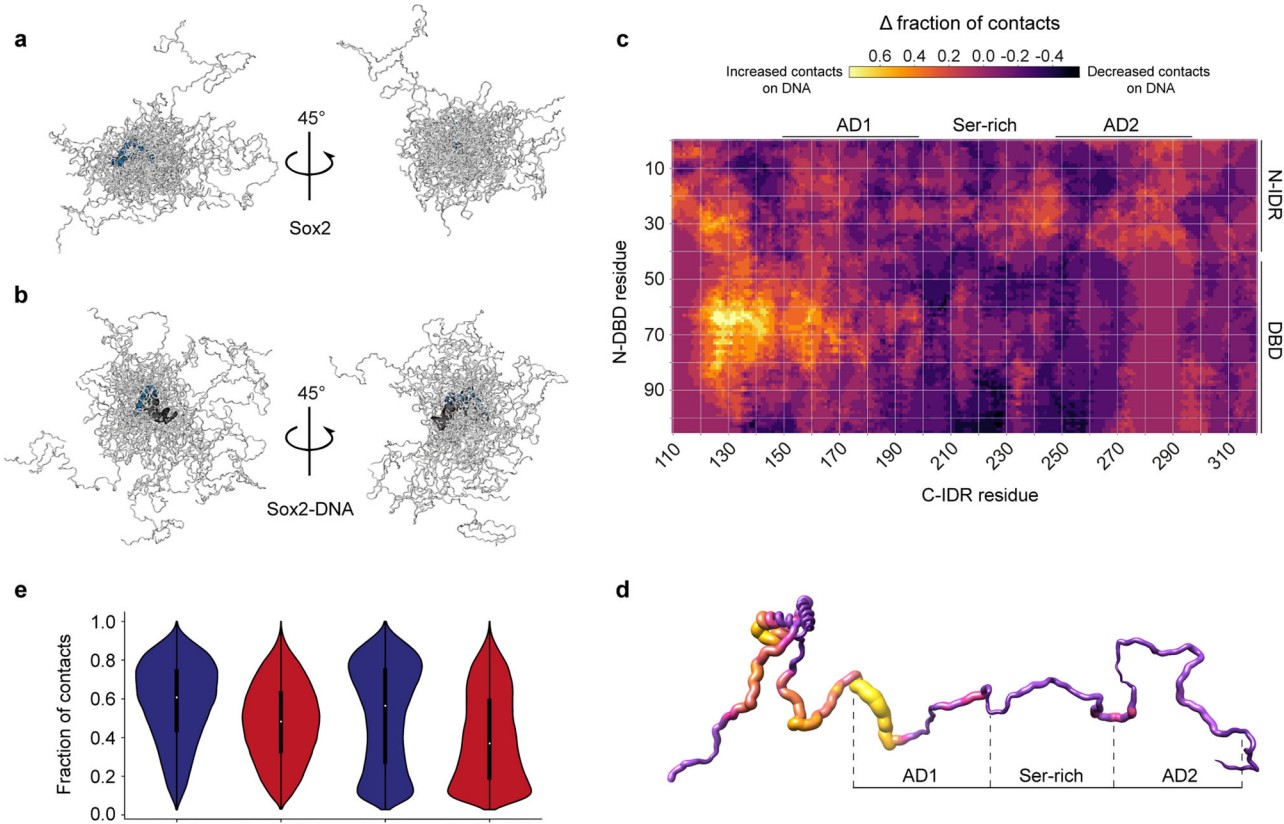

**Fig. 5 | Dynamic structural model of Sox2 ensembles, free and in complex with DNA.** 20 representative snapshots from the simulation for (**a**) free Sox2 and (**b**) DNA-bound Sox2. The DBD is shown in blue, C-IDR in light grey, DNA in dark grey. **c** Difference in the fraction of intramolecular contacts in Sox2 in the unbound and DNA-bound states. Regions showing positive values report on increased contacts in the bound state, whereas regions that have negative values have decreased contacts. **d** Difference in the fraction of intramolecular contacts in the unbound and DNA-bound states, projected on a schematic structure of Sox2. The colour scale is the same as in (**c**). **e** Violin plots showing the fraction of contacts for residues within AD1 and AD2 with the DBD. The whiskers encompass the difference between third and first percentile (inter-quartile range) and the white dots are the median values. Source data are provided as a Source Data file.

with an effect that is proximity-dependent and more pronounced for AD1 (Supplementary Fig. 10). When Sox2 binds to DNA, the C-IDR ensemble expands with more frequent excursions to extended states and thus a larger apparent $R_g$ (Fig. 5b). The difference in contacts between the DBD and C-IDR for free and DNA-bound Sox2 shows that the expansion observed experimentally upon DNA binding is coincident with an increased number of contacts between the N-IDR and C-IDR, and decreased overall contacts of both IDRs with the DBD (Fig. 5c). When Sox2 binds the DNA, the region experiencing the largest variation in contact space is the C-IDR AD1 (Fig. 5c, d, e) directly in line with significant DNA-induced CSPs in the AD1 region (Fig. 3h). Projecting the average number of contacts for each residue onto the Sox2 structure reveals an increase in proximity of regions overlapping with AD1 and DBD, when bound to DNA, but a decrease for AD2 (Fig. 5d). This effect is clearer when we plot the fraction of contacts specifically between the DBD and the two ADs in the free and DNA-bound states (Fig. 5e). On a residue-specific level, the difference in contacts between the C-IDR and DBD in the free and DNA bound states agrees reasonably well with the CSPs from our NMR experiments (Supplementary Fig. 11). We also analyzed whether loss of C-IDR contacts with the DBD might be accompanied by formation of new contacts with the DNA but there was no enrichment in contact formation beyond a short ~20-residue stretch immediately flanking the DBD which is known to stably bind into the DNA major groove[30,33] (Supplementary Fig. 11). Finally, the differential engagement of the two ADs is also re-iterated by analyzing the relaxation times of the contacts made by AD1 and AD2 with the DBD (Supplementary Fig. 11). The

correlation function of contact formation over time, fits better to a double exponential with distinct slow and fast components. The contact relaxation times for the two ADs are similar in the absence of DNA. However, in complex with DNA, the relaxation time for AD2 is reduced more than threefold compared to that of AD1 for which the contact lifetimes increases ($\tau_1^{AD1}/\tau_1^{AD2} = 0.8$, $\tau_2^{AD1}/\tau_2^{AD2} = 0.7$, for free Sox2; $\tau_1^{AD1}/\tau_1^{AD2} = 2.8$, $\tau_2^{AD1}/\tau_2^{AD2} = 4.7$ when bound to DNA) (Supplementary Table 5). These observations indicate that AD2 is accessible for a considerably longer time than AD1 when bound to DNA. Conversely, the serine-rich domain shows little difference in contact relaxation times before and after DNA binding (Supplementary Fig. 11).

## Discussion

It remains a major experimental and computational challenge to determine the conformational ensembles of disordered proteins and, as in the case of TFs, to relate them to function. This challenge is thoroughly exemplified by a lack of both entries in the protein data bank and confident AlphaFold prediction of full-length TFs. In our work, we have reconstructed a detailed, experimentally-driven description of the structural ensembles for both free full-length Sox2 and Sox2 in complex with DNA. The relatively low number of charges in the C-IDR render it a weak polyampholyte[40], which is expected to populate a rather collapsed structure. However, we found that the C-IDR engages in additional dynamic but weak interactions with the DBD, driven mainly by charge interactions between the two domains. Notably, the dimensions of Sox2 are very sensitive to salt in the range corresponding to physiological concentrations; local differences in

intracellular salt concentrations would be expected to further tune the accessibility of the C-IDR. This is a noteworthy observation: even though the charges in the C-IDR are relatively sparse and well distributed, their interactions with the DBD still confer a strong effect on the overall dimension of the protein. Keeping the C-IDR in a relatively compact state in the absence of DNA may be an evolved strategy to protect against unwanted interactions, premature degradation, aggregation, or condensate formation. Interactions with the DBD may also aid in keeping an otherwise aggregation prone C-IDR[47] soluble, until the right genomic binding site or coregulator is located. We found, in agreement with others[26,30], that DNA binding affinity was unaffected by interdomain interactions; sustaining sufficiently weak interactions that maintain the advantages of a highly dynamic ensemble may be crucial to modulate the accessibility of the ADs without disturbing DNA binding. Intramolecular interactions for other nucleic acid binding proteins have been reported to influence binding affinity, mostly through interactions mediated by strongly charged but short regions[11-13]. Other TFs have recently been reported to have similar interdomain interactions, including cases such as B-MYB where a short and strongly positively charged region interacted with the DBD but with little effects on DNA binding affinity[48]. The TFs p53 and MYC/MAX have also been demonstrated to partake in intramolecular electrostatic interactions with their DBDs, to a degree dependent on their phosphorylation state[38,49,50]. For p53, interdomain interactions had no effect on binding affinity to specific DNA but led to a 5-fold affinity reduction to non-specific DNA, thus increasing specificity, a scenario not recapitulated by Sox2. Sox2 has several phosphorylation sites in the C-IDR, which may enable tuning of DNA binding affinity or specificity[51]. For example, phosphorylation of Thr116, adjacent to the DNA binding HMG box, has been shown to be necessary for recruitment to certain stem-cell dependent promoters[52]. Given the relatively few charged residues in the C-IDR of Sox2, a single PTM that affects the charge state might have a large effect on the magnitude of interdomain interactions, potentially leading to ultra-sensitivity in IDR dimensions and thus immediate shaping of the Sox2 interactome.

Our CG model shows that when using interdomain contacts and overall dimensions as an indirect proxy for accessibility, we observe changes in accessibility upon binding DNA which localise largely to regions overlapping with the ADs, harbouring many charged residues. Interestingly, part of AD1 shows decreased accessibility upon DNA binding (Fig. 5c, d, e) whereas much of the remainder of the C-IDR, including AD2, has more than fourfold increased accessibility when viewed through the lens of relaxation times. Even though the precise boundaries of ADs remain to be defined, our results show variable responses of discrete C-IDR regions to DNA binding. It is likely that a combination of residue proximity to the DBD and DNA, charge number, and charge distribution[40] will dictate the exact conformational pattern for specific TFs, but deciphering the details of that code is an important future task. Addition of negative charges, e.g., in the form of phosphorylations, might be expected to enhance the interaction with the positively charged DBD and thus increase occupancy in a compact ensemble, rendering ADs more or less accessible to coregulators dependent on the sequence position of the negative charge. For example, there is a conserved positive region flanking the DBD on the C-terminal side that we observe to stably interact with the DNA in our simulations, in agreement with previous studies[30]. Phosphorylations in this region (e.g., on Thr116[53]) would be expected to decrease interactions with DNA, potentially increasing the accessibility of AD1, while a phosphorylation further downstream in the sequence (e.g., Ser251) might increase interactions with the DBD leading to decreased accessibility of AD2. Increased interactions between the C-IDR and DBD might in some cases lead to less efficient DNA binding, which could explain why Sox2 binds certain enhancers less when phosphorylated in Ser251 which is close to AD2[54]. Nonetheless, PTM effects are complex and more intricate than simple modulation of interdomain interaction

strength. Generally, our structural model of Sox2 will aid in rationalising the effects of PTMs as well as linking them to conformational changes and cofactor binding.

The pioneer activity of Sox2 is dependent on its ability to bind to and alter the structure of nucleosomes[16,32,33]. Upon binding nucleosomes, the Sox2 C-IDR goes through similar, albeit not identical, conformational rearrangements to those that follow its binding to short DNA, suggesting that our reconstructed ensemble will also be generally populated on nucleosomes. Fluorescence lifetime analysis in the nucleosome-bound state showed that the C-IDR is slightly less dynamic on the submillisecond timescale when compared with the DNA-bound state; whether this is due to steric restrictions in the local conformational space or due to a direct interaction with the core histones is currently unknown. Nonetheless, a compelling hypothesis is that the exact nature of the Sox2 binding site, i.e., whether it is on free DNA or in different locations on a nucleosome particle, will dictate the degree of AD accessibility and the resulting interaction profile. Binding experiments with interaction partners are needed to reveal whether that is a feasible model but it would offer possibilities for specifically targeting interactions with nucleosome-bound Sox2 while excluding those that involve accessible DNA. Our structural model is a first step in that direction and creates a platform for mapping the effects of mutations, environment, and binding partners on the structural ensemble of the Sox2 IDR.

Members of the SoxB family of TFs (Sox1, Sox2, and Sox3) share general composition features in their IDRs, such as the number and position of charges, and therefore the conformational dynamics that we observe for Sox2 are likely to be closely applicable to this family (Supplementary Fig. 12). Beyond the SoxB family, these types of interdomain interactions may be very common among TFs to restrain and finely tune the accessibility of ADs to varying degrees before and after they have located their binding sites. In fact, AlphaFold predictions and bioinformatics analysis support that most TFs share a similar architecture and charge profile (positively charged DBD, modest numbers of charges in IDRs)[2]. Further studies will reveal whether the accessibility tuning modulates the interaction equilibrium of TFs with coactivators within the transcriptional machinery. Finally, this type of ensemble redistribution with expansion excursions on DNA may also be linked to condensate formation, which has been suggested to be involved in transcriptional regulation, potentially rendering phase separation more likely to occur once TFs have located their DNA or nucleosome targets.

## Methods
### Protein expression and purification
The DNA coding for all Sox2 constructs was inserted into a modified pET24b vector. The vector contains codes for a hexahistidine small ubiquitin-like modifier (His$_6$-SUMO) tag added to the N-terminal of the constructs. Mutants were made using the QuikChange Lightning kit from Agilent using primers from Integrated DNA Technologies (IDT). All constructs were expressed in Lemo21(DE3) cells (New England BioLabs) cultured in LB-broth medium, or M9 minimal medium containing $^{15}$N-NH$_4$Cl or $^{15}$N-NH$_4$Cl and $^{13}$C$_6$-glucose. Expression was induced at OD$_{600}$ 0.5–0.7 with 0.4 mM Isopropyl β- d-1-thiogalactopyranoside (IPTG) and cells were grown for 2–3 h at 37 °C with vigorous shaking. Cells were harvested by centrifugation at $4500 \times g$ for 15 min and resuspended in Buffer A (50 mM NaH$_2$PO$_4$, 300 mM NaCl, 10 mM imidazole, 6 M urea, 1 mM dithiothreitol (DTT), pH 8.0) for overnight lysis at 4 °C. The soluble fraction was collected by centrifugation at $40,000 \times g$ for 1 h at 4 °C and loaded onto a 5 ml HisTrap HP column (Cytiva) equilibrated with Buffer A. The column was washed with 10 column volumes (CV) of Buffer A and eluted with Buffer A with imidazole concentration adjusted to 500 mM. Eluted samples were dialysed overnight against Buffer B

(50 mM Tris, 150 mM NaCl, 1 mM DTT, pH 8.0), followed by ULP1 protease (made in-house) cleavage to remove the His$_6$-SUMO tag. Following cleavage, the N-DBD and DBD constructs were dialysed against Buffer C (50 mM NaH$_2$PO$_4$, 6 M Urea, pH 8.0) overnight and loaded onto a 5 ml HiTrap SP Sepharose FF column (Cytiva). The SUMO tag eluted during the 10 CV wash step (Buffer C) and the proteins were eluted with Buffer C with NaCl concentration adjusted to 500 mM. Full-length Sox2 and C-IDR precipitated from solution following the removal of the His$_6$-SUMO tag, the precipitate was recovered by centrifugation and resuspended in Buffer C. All protein preparations were concentrated using Amicon Ultracentrifugal filters (Merck), reduced with DTT and purified by reversed-phase high-performance liquid chromatography (RP-HPLC) using a ZORBAX 300SB-C3 column (Agilent) with flow rate of 2.5 ml/min starting at 95% RP-HPLC solvent A 99.9% H$_2$O, 0.1% trifluoroacetic acid (TFA)(Sigma) and 5% RP-HPLC solvent B (99.9% acetonitrile, 0.1% TFA) and going to 100% RP-HPLC solvent B over 95 min. Protein purity was analysed by SDS-PAGE, identity confirmed by mass spectrometry, and samples were lyophilised and stored at −20 °C.

### Protein labelling

Lyophilised proteins were resuspended in labelling buffer (0.1 M potassium phosphate, 1 M urea, pH 7.0) and labelled overnight at 4 °C using Cy3B maleimide (donor) (Cytiva) (0.7:1 dye to protein ratio). The reaction was quenched using DTT and RP-HPLC was then used to remove unreacted dye, and separate unlabelled and double donor-labelled proteins. The proteins were lyophilised overnight, then resuspended in labelling buffer and labelled overnight at 4 °C using CF660R maleimide (acceptor) (Sigma). The reaction was quenched using DTT and RP-HPLC was then used to separate donor-donor doubly labelled and acceptor-acceptor doubly labelled proteins. Donor-acceptor labelled proteins were lyophilised, resuspended in 8 M GdmCl, frozen in liquid N$_2$, and stored at −80 °C.

### DNA labelling

Aliquots of 5–10 nmol oligonucleotide (oligonucleotides contained a thymine modified with a C6-amino linker for the reaction with the NHS ester of the dyes) (IDT) were dissolved in 50 μl DNA labelling buffer (0.1 M sodium bicarbonate, pH 8.3) and labelled with either Cy3B NHS ester (Cytiva) or CF660R NHS ester (Sigma) in a 2:1 dye to DNA ratio. The reaction was incubated for at least two hours at room temperature, then ethanol precipitated to remove excess dye. Pellet was redissolved in 100 μl of 95% RP-HPLC solvent C (0.1 M triethylammonium acetate) and 5% RP-HPLC solvent D (acetonitrile) and separated from the unreacted dye and unlabelled oligonucleotide with RP-HPLC using a ReproSil Gold 200 C18 column (Dr. Maisch), labelled oligonucleotides were collected and lyophilised. Oligonucleotides intended for PCR amplification were resuspended in double distilled water (ddH$_2$O) to a final concentration of 2.5 μM and stored at −20 °C. Oligonucleotides intended for smFRET measurements were resuspended in DNA annealing buffer (10 mM Tris, 50 mM NaCl, 1 mM EDTA, pH 7.5) and mixed with equimolar amounts of the reverse compliment oligonucleotide labelled with either Cy3B or CF660R. Sample was placed on a heating block at 95 °C for 5 min, heating was turned off and samples allowed to cool slowly to room temperature to anneal the donor labelled and the acceptor labelled oligonucleotide strands. Labelled DNA was aliquoted, frozen in liquid N$_2$, and stored at −80 °C.

### Nucleosome reconstitution

PCR amplification of a pJ201 plasmid containing the 147 bp Widom sequence was used to generate DNA for nucleosome reconstitution. The amplification took place using either fluorescently labelled oligonucleotides (see DNA labelling) or unlabelled oligonucleotides (IDT). The oligonucleotides were designed to insert a Sox2 binding site

(CTTTGTTATGCAAAT) and to extend the 147 bp Widom sequence by 25 bp linkers on either side. The PCR reactions were ethanol precipitated before being purified using a DNA Clean and Concentrator Kit (Zymo Research). The concentration of the DNA was determined by UV Vis. For the list of primer and DNA sequences see Supplementary Table 3. To reconstitute nucleosomes 10 pmol of purified 197 bp Widom sequence containing a Sox2 binding site were used. The DNA was mixed with 1.0–1.75 molar equivalents of recombinant core histone octamer (The Histone Source) in 10 mM Tris, 0.1 mM EDTA, 2 M KCl, pH 7.5, on ice. The reaction was then transferred to a Slide-A-Lyzer MINI dialysis button (Thermo Fisher Scientific) and dialysed against a linear gradient of 10 mM Tris, 0.1 mM EDTA, 10 mM KCl, pH 7.5 over 20 h at 4 °C. Constant volume of buffer was maintained by removing buffer at the same rate as fresh buffer with 10 mM KCl was added using a peristaltic pump. Samples were transferred to microcentrifuge tubes and centrifuged for 5 min at 20.000 × g, 4 °C to remove aggregates, supernatant was collected. Concentration was determined via absorbance at 260 nm and 0.5 pmol of the reaction was loaded on a 0.7% agarose gel and run for 90 min at 90 V with 0.25 × Tris-borate as running buffer. Following staining with GelRed (Biotium) gels were imaged using Gel Doc EZ gel system (Bio-Rad). Only samples that contained <5% free DNA were used for measurements.

### Single-molecule spectroscopy

All single molecule fluorescence experiments were conducted at 23 °C using a MicroTime 200 (PicoQuant) connected to an Olympus IX73 inverted microscope. The donor dye was excited using a 520 nm diode laser (LDH-D-C-520, PicoQuant) using pulsed interleaved excitation[55](PIE) with a 640 nm diode laser (LDH-D-C-640, PicoQuant) to alternate excitation of donor and acceptor dyes with a repetition rate of 40 MHz. The laser intensities were adjusted to 40 μW at 520 nm and 20 μW at 640 nm (PM100D, Thorlabs). Excitation and emission light was focused and collected using 60 × water objective (UPLSA-PO60XW, Olympus). Emitted fluorescence was focused through a 100 μm pinhole before being separated first by polarisation and then by donor (582/64 BrightLine HC, Semrock) and acceptor (690/70 H Bandpass, AHF) emission wavelengths, into four detection channels. Detection of photons took place using single photon avalanche diodes (SPCM-AQRG-TR, Excelitas Technologies). The arrival time of detected photons was recorded with a MultiHarp 150 P time-correlated single photon counting (TCSPC) module (PicoQuant). All experiments were performed in μ-Slide sample chambers (Ibidi) at RT in TEK buffer (10 mM Tris, 0.1 mM EDTA, pH 7.4) with varying KCl concentrations. For photoprotection 143 mM 2-mercaptoethanol (Sigma) was added, along with 0.01% (v/v) Tween-20 (AppliChem) to reduce surface adhesion. In experiments using denaturants, the exact concentration of denaturant was determined from measurement of the solution refractive index[56].

### Analysis of transfer efficiency histograms

Data for transfer efficiency histograms were collected from 50 to 100 pM of freely diffusing double labelled Sox2, DNA or nucleosomes. All data was analysed using the Mathematica scripting package "Fretica" (https://schuler.bioc.uzh.ch/programs/) developed by Daniel Nettels and Ben Schuler. Fluorescence bursts were first identified by combining all detected photons with less than 100 μs interphoton times. Transfer efficiencies within each fluorescence burst were calculated according to $E = n'_A/(n'_A + n'_D)$, where $n'_A$ and $n'_D$ are the number of acceptor and donor photons, respectively. The number of photons were corrected for background, direct acceptor excitation, channel crosstalk, differences in dye quantum yields and photon detection efficiencies[36]. The resulting bursts were then filtered to remove bursts where the acceptor bleaches during the transit of the molecule through the confocal volume[57], which otherwise can cause a bias towards lower FRET. Occasional fluorescence bursts with photon

counts more than three times higher than the mean signal binned at 1 s, corresponding to aggregates, were removed before data analysis. The labelling stoichiometry ratio (S) was determined according to:

$$S = \frac{n_D^D + n_A^D}{n_D^D + n_A^D + n_A^A} \qquad (1)$$

where $n_{D/A}^D$ is the number of detected donor or acceptor photons after donor excitation and $n_A^A$ is the number of detected acceptor photons after acceptor excitation. To construct the final transfer efficiency histograms, we selected bursts that have $S = 0.3 - 0.7$ which allowed us to filter out bursts that originate from molecules that lack an active acceptor. In some cases, a large donor-only population can cause residual donor-only bursts to remain after filtering.

To extract mean FRET efficiencies, the histograms were fitted to an appropriate number of Gaussian or logNormal distribution function, corresponding to one or more populations. Multiple transfer efficiency histograms for binding affinity analysis were fitted globally, where some parameters were shared across different measurements. For distance calculations based on the transfer efficiencies for DNA and nucleosomes the Förster equation

$$E(r) = \frac{1}{1 + r^6/R_0^6} \qquad (2)$$

was used with $R_0 = 6.0$ nm for a Cy3B/CF660R dye pair. For double labelled proteins involving disordered segments we converted mean transfer efficiencies $\langle E \rangle$ to root-mean-square end-to-end distances $R = \sqrt{\langle r^2 \rangle}$ by numerically solving the following transcendental equation:

$$\langle E \rangle = \int_0^\infty dr\, E(r) P(r) \qquad (3)$$

Here, $P(r)$ denotes the distance probability density function of the SAW-$\nu$ model[58], given by

$$P(r) = A \frac{4\pi}{R} \left(\frac{r}{R}\right)^{2+(\gamma-1)/\nu} \exp\left(-\alpha\left(\frac{r}{R}\right)^{1/(1-\nu)}\right), \qquad (4)$$

which is characterised by the critical exponents $\nu$ and $\gamma \approx 1.1615$. The constants $A$ and $\alpha$ are determined by requiring $P(r)$ to be normalised and to satisfy $\langle r^2 \rangle = R^2$, respectively. The dependency on $\nu$ in $P(r)$ is removed by assuming that a scaling law $R = bN^\nu$ must hold and substituting $\nu = \ln\left(\frac{R}{b}\right)/\ln(N)$ into the expression for $P(r)$, where $b \approx 0.55$ nm for proteins and $N$ denotes the number of monomers between the fluorescent groups. The associated radius of gyration $R_g$ can be approximated as

$$R_g \approx R \sqrt{\frac{\gamma(\gamma+1)}{2(\gamma+2\nu)(\gamma+2\nu+1)}}. \qquad (5)$$

In denaturation experiments, the Förster radius was corrected for changes in refractive index according to[41]:

$$R_0^6(c_D) = R_{0,0}^6 \left(\frac{nR_0^6}{n(c_D)}\right)^4 \qquad (6)$$

where $n(c_D)$ denotes the refractive index of the sample at denaturant concentration $c_D$.

Fluorescence anisotropy values were determined for fluorescently labelled variants using polarisation-sensitive detection in the single-molecule instrument[59], and were between 0.04 and 0.14 both for the monomeric proteins and the proteins in complex with DNA, indicating sufficiently rapid orientational averaging of the fluorophores to justify the approximation $\kappa^2 \approx 2/3$ used in Förster theory[60].

## Fluorescence correlation spectroscopy

To determine the diffusion time of labelled Sox2, we performed fluorescence correlation spectroscopy by correlating the intensity fluctuations in fluorescence in an smFRET experiment according to

$$G_{ij}(\tau) = \frac{\langle \delta n_i(0) \delta n_j(\tau) \rangle}{\langle n_i \rangle^2} \qquad (7)$$

where $i,j = A, D$ and $n_i(0)$ and $n_j(\tau)$ are fluorescence count rates for channels $i$ and $j$ at time 0 and after a lag time t, respectively, and $\delta n_{i,j} = n_{i,j} - \langle n_{i,j} \rangle$ are the corresponding deviations from the mean count rates.

Data for nsFCS[61] were collected using continuous-wave excitation at 520 nm and a ~100-pM sample of double-labelled free Sox2, DNA-bound Sox2, or isolated C-IDR. Donor and acceptor fluorescence photons from only the FRET subpopulation were used for correlations at 1 ns binning time. Photons were cross-correlated between detectors to avoid the effects of detector dead times and after-pulsing on the correlation functions. Cross-correlation curves between acceptor and donor channels were fit and analyzed as described previously[6]. Briefly, the correlation curves were fit over lag time interval from −1 μs to +1 μs using

$$g_{ij}(\tau) = a \left(1 - c_{ab} e^{\frac{-|\tau|}{\tau_{ab}}}\right)\left(1 + c_{cd} e^{\frac{-|\tau|}{\tau_{cd}}}\right), \qquad (8)$$

where $i$ and $j$ indicate donor ($D$) or acceptor ($A$) fluorescence emission; the amplitude $a$ depends on the effective mean number of molecules in the confocal volume and on the background signal; $c_{ab}$, $\tau_{ab}$, $c_{cd}$ and $\tau_{cd}$ are the amplitudes and time constants of photon antibunching (ab) and chain dynamics (cd), respectively. $\tau_{cd}$ can be converted to the reconfiguration time of the chain, $\tau_r$, by assuming that the chain dynamics can be modelled as a diffusive process in the potential of mean force derived from the sampled inter-dye distance distribution $P(r)$[61,62] based on the SAW-ν model[36,58]. The correlation functions are displayed with a normalisation to 1 at their respective values at 0.5 μs.

## Fluorescence lifetime analysis

Fluorescence lifetimes were estimated from the mean donor detection times $\langle t_D \rangle$ after their respective excitation pulse. The fluorescence lifetimes were then plotted against corresponding transfer efficiencies in two-dimensional scatter plots, where $\tau_{DA}/\tau_D = \langle t_D \rangle/\tau_D$ was calculated for each burst for an intrinsic donor lifetime $\tau_D$. For a fixed distance between the donor and acceptor, the ratio $\langle \tau_{DA} \rangle/\tau_D$ must equal $1 - E$ (Fig. 1c, diagonal line), whereas for systems that rapidly sample a broad distance distribution this ratio significantly deviates from $1 - E$. For a rapidly fluctuating distance described by a probability density function $P(r)$ of the interdye distance $r$, the distribution of distances affects the average fluorescence lifetime $\langle \tau_{DA} \rangle$ according to

$$\frac{\langle \tau_{DA} \rangle}{\tau_D} = 1 - \langle E \rangle + \frac{\sigma^2}{1 - \langle E \rangle} \qquad (9)$$

Here, the variance $\sigma^2$ is given by

$$\sigma^2 = \langle E^2 \rangle - \langle E \rangle^2 = \int_0^\infty dr\, [E(r) - \langle E \rangle]^2 P(r). \qquad (10)$$

## Determination of denaturant association coefficients

Association constants ($K_a$) of GdmCl and urea were determined using a weak denaturant binding model[63,64] with the form

$$E(c_D) = \frac{E_0 + \Delta E \, K_a \, c_D}{1 + K_a \, c_D} \tag{11}$$

where $c_D$ is the denaturant concentration, with $K_a$, $\Delta E$, and $E_0$ being fit parameters.

## Binding affinity measurements

Transfer efficiency histograms were recorded for either double labelled Sox2 or DNA with increasing concentration of unlabelled binding partner until the transfer efficiency remained stable. Gaussian peak functions were used to fit the histograms into two subpopulations, bound and unbound. From the relative areas of these subpopulations the fraction of bound species ($\theta$) could be quantified. To aquire the dissociation constant ($K_D$) a binding isotherm was fit using

$$\theta = \frac{c_{X,tot} + K_D + c_{Y,tot} + \sqrt{(c_{X,tot} + K_D + c_{Y,tot})^2 - 4c_{X,tot}c_{Y,tot}}}{2c_{Y,tot}} \tag{12}$$

where $c_{X,tot}$ and $c_{Y,tot}$ are the total concentrations of Sox2 or DNA, depending on which molecule is kept at a constant concentration.

## CD spectroscopy

Far-UV CD spectra were recorded on a Jasco J-1100. All spectra were recorded at 25 °C in 25 mM NaH$_2$PO$_4$, 25 mM NaCl at pH 8.0 using a 1 mm cuvette. Spectra were recorded between 250 and 190 nm, data pitch was 0.1 nm, digital integration time of 0.25 s, scan speed 20 nm/min and accumulating 3 scans. Protein concentrations ranged from 2 to 5 µM. Identical measurements were taken of the buffer, which was then subtracted from the measurements. The ellipticity was converted to mean residual ellipticity using

$$MRE = \frac{mdeg}{10 \times L \times C \times N}, \tag{13}$$

where $L$ is the path length in cm, $C$ is the concentration in molar, and $N$ is the number of peptide bonds.

## NMR spectroscopy

All NMR spectra were recorded on a Bruker Avance Neo 800 MHz spectrometer or Avance III HD 750 MHz spectrometer equipped cryogenic probe. Samples were recorded in 20 mM NaH$_2$PO$_4$, 50 mM NaCl, 5 mM DTT, 125 µM DSS, 5% D$_2$O (v/v) at pH 5.5 and 15 °C to minimise amide exchange. The raw free induction decays (FIDs) were transformed using NMRPipe[65] and analysed using CcpNmr software[66]. Backbone nuclei of $^{13}$C,$^{15}$N-labelled Sox2 were assigned in the unbound state (110 µM $^{13}$C-$^{15}$N-labelled Sox2 from analysis of $^1$H$^{15}$N HSQC, HNCACB, CBCA(CO)NH, HN(CO)CA, HNCO, and HN(CA)NNH multi-dimensional NMR spectra (BMRB accession number 51964)). The intensity of backbone resonances from the DBD were too weak in full-length Sox2 for direct assignments but could be transferred from assignments of the isolated N-DBD (Fig. 2 and Supplementary Fig. 1). Secondary structure content in Sox2 was determined from secondary C$^\alpha$ chemical shifts using a random coil reference for intrinsically disordered proteins[67].

$T_1$ and $T_2$ $^{15}$N relaxation times were determined from 2 × 2 series of $^1$H$^{15}$N HSQC spectra with varying relaxation delays and using pulsed-field gradients for suppression of solvent resonances. The series were recorded at 800 MHz (1H), using 8 (20 ms, 60 ms, 100 ms, 200 ms, 400 ms, 600 ms, 800 ms and 1200 ms) and 8 (0 ms, 33.9 ms, 67.8 ms, 101.8 ms, 135.7 ms, 169.6 ms, 203.5 ms and 271.4 ms) different

relaxation delays for $T_1$ and $T_2$, respectively. CcpNmr Analysis software[66] was used to fit the relaxation decays to single exponentials and determine relaxation times.

Binding induced weighted CSPs were measured at a protein concentration of 30 µM in absence and presence of a 1.1 fold excess unlabelled double-stranded DNA with Sox2 binding sequence (Supplementary Table 3). CSPs were calculated as[68]

$$CSP = \sqrt{\frac{1}{2}\left((\Delta\delta_H)^2 + (0.154^*\Delta\delta_N)^2\right)} \tag{14}$$

The dissociation constant $K_D$ for DBD/C-IDR interactions was quantified using chemical shift perturbation analysis[69], by employing the observed chemical shift changes ($\Delta\delta_{obs}$) of $^{15}$N-labelled C-IDR upon the addition of unlabelled N-DBD. $K_D$ was calculated using:

$$\Delta\delta_{obs} = \Delta\delta_{max}\left\{([P]_t + [L]_t + K_D) - \left[([P]_t + [L]_t + K_D)^2 - 4[P]_t[L]_t\right]^{1/2}\right\}/2[P]_t \tag{15}$$

where $\Delta\delta_{obs}$ is the observed chemical shift change, $\Delta\delta_{max}$ is the maximum chemical shift change, $[P]_t$ is the total C-IDR concentration, $[L]_t$ is the total N-DBD concentration[70]. The resulting chemical shift changes were fitted to the formula and $K_D$ and $\Delta\delta_{max}$ were determined.

## Simulations

**Protein model.** The all-atom starting structure for Sox2 was obtained from the electron microscopy structure of Sox2 bound to a nucleosome (PDB: 6T7B[33]). The disordered regions, not available in the starting structure, were modelled using the modeller plugin[71] embedded in UCSF Chimera[72]. Each residue of Sox2 was represented as a single bead mapped to the C$^\alpha$ atom of the starting full-atom structure. The simulation parameters used in the current work are identical to those outlined in ref. 44. We used the following potential energy function describing protein-protein interactions[6]:

$$\begin{aligned}V_{protein-protein} &= \frac{1}{2}\sum_{i=1}^{N} k_b\left(d_i - d_i^0\right)^2 + \frac{1}{2}\sum_{i=1}^{N} k_\theta\left(\theta_i - \theta_i^0\right)^2 \\ &+ \sum_{i=1}^{N-2}\sum_{m=1}^{4} k_{i,m}\left(1 + \cos(n\phi_i - \delta_{i,m})\right) + \sum_{i<j}\frac{q_i q_j}{4\pi\epsilon_d\epsilon_0 d_{ij}}e^{-\frac{d_{ij}}{\lambda_D}} \\ &+ \sum_{(i,j)\in Native}\varepsilon_{ij}\left[13\left(\frac{\sigma_{ij}}{d_{ij}}\right)^{12} - 18\left(\frac{\sigma_{ij}}{d_{ij}}\right)^{10} + 4\left(\frac{\sigma_{ij}}{d_{ij}}\right)^6\right] \\ &+ \sum_{(i,j)\notin Native}4\varepsilon_{pp}\left[\left(\frac{\sigma_{ij}}{d_{ij}}\right)^{12} - \left(\frac{\sigma_{ij}}{d_{ij}}\right)^6\right]\end{aligned} \tag{16}$$

where the first three terms describe bonded while the second three non-bonded interactions. Bonds and angles (first and second terms, respectively) are treated with harmonic potentials with force constants $k_b$, $k_\theta$ for bond lengths and equilibrium values $d_i^0$ and $\theta_i^0$ for angles. Both assignments are based on the distances and angles between the C$^\alpha$ atoms in the all-atom starting structure. A cosine-based dihedral potential (third term) was used to sample the behaviour of four beads linked by three bonds, with the dihedral angle parameters described by the force constant $k_{i,m}$ and a phase shift term $\delta_{i,m}$. These parameters are obtained from a sequence-specific dihedral potential, informed by structures deposited in the RCSB[73]. Electrostatic interactions are described in the fourth term using a screened Coulomb potential. While lysine and arginine are assigned a charge of +1, aspartate and glutamate are given a charge of −1 and histidine a charge of +0.5, considering that the imidazole side chain in histidine usually holds a p$K_a$ of ~6.0. The charge of all the other beads was set to 0. The Coulomb term is composed of terms pertaining to the charge of a

residue ($q_i$), the dielectric constant of water ($\epsilon_d$) set to a value of 80, the permittivity of the medium ($\epsilon_0$), and the Debye screening length ($\lambda_D$), which is given by:

$$\lambda_D = \frac{\epsilon_0 \epsilon_d k_B T}{2 N_A e^2 I},$$ (17)

where the Boltzmann constant and temperature are described by $k_B$ and $T$, respectively, in addition to the Avogadro's number $N_A$, the elementary charge $e$ and the ionic strength $I$. As such, different ionic strength values were mimicked by altering the Debye screening length. The fifth and sixth terms collectively describe short-range attractive interactions between beads, separated by a distance $d_{ij}$. Native interactions pertain to the folded domains and are computed using a 12−10−6 pair potential, which has been successfully adopted in a Gō-model employed to investigate protein folding by Karanikolas and Brooks[73]. In this potential, $\varepsilon_{ij}$ describes the strength of the interaction calculated in accordance with a native-centric model[73], with $\sigma_{ij} = \frac{(\sigma_i + \sigma_j)}{2}$, determined based on $C^\alpha$-$C^\alpha$ distances in the crystal structure. Conversely, the interaction between residues located in the disordered regions and between disordered regions and the folded Sox2 domain, is described by a simpler 12-6 Lennard-Jones potential with $\varepsilon_{pp}$ set to a value of 0.16 $k_B T$ (-0.4 kJ mol$^{-1}$) and $\sigma_{ij}$ to a fixed value of 0.6 nm. These values have previously been effective in giving the best agreement with experientially derived FRET efficiencies[6,44].

## DNA model
The CG representation used for the DNA is comprised of three beads representing the phosphate, ribose and base moieties of nucleic acids, mapped to the P, C4' and N1 atoms in the all-atom DNA structure, respectively. All phosphate beads were assigned a charge of −1, while ribose and base beads were not charged. Initially, to obtain a reliable model of the Sox2-DNA binding site, a segment of the nucleosome containing the Sox2 consensus sequence with Sox2 bound to it was taken from the electron microscopy structure with accession code 6T7B[33]. This fragment of DNA was then mutated to match the DNA sequence used in experiments. This modelling strategy ensured a lower strain between the bound Sox2 and the segment of DNA, which would have otherwise been modelled as a straight DNA segment, while Sox2 preferentially binds to curved, nucleosomal DNA.

The interactions between DNA beads are given by the following potential energy function:

$$V_{DNA} = \frac{1}{2}\sum_{i<N} k_b (d_i - d_i^0)^2 + \frac{1}{2}\sum_{i<N-1} k_\theta (\theta_i - \theta_i^0)^2 + \sum_{i<j}\frac{q_i q_j}{4\pi\epsilon_d\epsilon_0 d_{ij}} e^{-\frac{d_{ij}}{\lambda_D}}$$
$$+ \sum_{(i,j)\in\text{stack}} 4\varepsilon_{stack}\left[\left(\frac{\sigma_{ij}}{d_{ij}}\right)^{12} - \left(\frac{\sigma_{ij}}{d_{ij}}\right)^6\right]$$
$$+ \sum_{(i,j)\in\text{pair}} 4\varepsilon_{pair}\left[\left(\frac{\sigma_{ij}}{d_{ij}}\right)^{12} - \left(\frac{\sigma_{ij}}{d_{ij}}\right)^6\right]$$
$$+ \sum_{(i,j)\notin\text{pair,stack}} 4\varepsilon_{ns}\left[\left(\frac{\sigma_{ij}}{d_{ij}}\right)^{12} - \left(\frac{\sigma_{ij}}{d_{ij}}\right)^6\right]$$ (18)

The bond distances and bond angles between the beads are described in the first and second terms, respectively. These two terms are identical to the ones used to treat the same interactions for protein beads. Specific values for these parameters were chosen to reproduce a typical persistence length (~50 nm) of double helical DNA at the equilibrium[74]. As described above, a modified Coulomb potential where $\epsilon_0$ is the permittivity of free space and $\epsilon_d$ a dielectric constant set to 80 to mimic an aqueous environment, was used to account for electrostatic interactions, with a screening between charged beads

defined by the Debye length $\lambda_D$. Non-bonded intearactions between DNA beads were accounted for by stacking and pairing terms and described by Lennard-Jones potentials, as shown by the third and fourth terms, respectively. While $\varepsilon_{stack}$ was set to 3.0 $k_B T$ (-7.5 kJ mol$^{-1}$), $\varepsilon_{pair}$ was set to 3.5 $k_B T$ (-8.8 kJ mol$^{-1}$), consistent with previously estimated free energy values for base stacking and shown to effectively reproduce the dynamics of B-DNA[44]. For DNA beads not involved in stacking or pairing interactions, a weakly attractive potential of $\varepsilon_{ns} = 0.04 k_B T$ (-0.10 kJ mol$^{-1}$) was employed. All values used to parameterise the DNA in the current work have been previously used to succesfully describe DNA in a protein-DNA complex[44].

## Protein-DNA interaction potential
The potential energy describing the interactions between protein and DNA beads has the following functional form:

$$V_{protein-DNA} = \sum_{i<j}\frac{q_i q_j}{4\pi\epsilon_d\epsilon_0 d_{ij}} e^{-\frac{d_{ij}}{\lambda_D}} + \sum_{(i,j)\in\text{native}} \varepsilon_{ij}\left[13\left(\frac{\sigma_{ij}}{d_{ij}}\right)^{12} - 18\left(\frac{\sigma_{ij}}{d_{ij}}\right)^{10} + 4\left(\frac{\sigma_{ij}}{d_{ij}}\right)^6\right]$$
$$+ \sum_{(i,j)\in\text{native}} 4\varepsilon_{pd}\left[\left(\frac{\sigma_{ij}}{d_{ij}}\right)^{12} - \left(\frac{\sigma_{ij}}{d_{ij}}\right)^6\right].$$ (19)

While native contacts describe interactions between DNA and Sox2 DBD, non-native contacts address interactions between DNA and the disordered tail regions of Sox2. Native contacts between bead pairs were identified from the all-atom starting structure, using a cutoff-based analysis of the crystal structure of the Sox DBD-DNA complex (PDB accession code: 6T7B[33]). If the distance between any atom of a protein residue and any atom of a nucleotide would fall below 0.5 nm, the contact was considered native and the strength of the interaction $\varepsilon_{ij}$ would be set to 2 $k_B T$ (-5 kJ mol$^{-1}$). Otherwise the contact was considered non-native and $\varepsilon_{pd}$ was set to 0.06 $k_B T$ (-0.15 kJ mol$^{-1}$). For all contacts, $\sigma_{ij}$ was set to a value of 0.5 nm. The values of $\varepsilon_{ij}$ and $\varepsilon_{pd}$ have previously been optimised to yield the best agreement between experimental and simulated FRET efficiencies[44].

## Langevin dynamics simulations of the protein and protein-DNA complexes
Langevin dynamics simulations of the protein in isolation and bound to DNA were performed using GROMACS version 5.1.4[75]. Each system was placed at the centre of a cubic box measuring 30 and 120 nm$^3$ for the proteins and protein-DNA complexes, respectively. All simulations were performed using periodic boundary conditions and charge screening was obtained considering the effect of monovalent salt at concentrations ranging from 40 to 800 mM, mimicked by adjusting the Debye length $\lambda_D$. After energy minimisation, each system was simulated for a total of 20 μs (4 replicates of 5 μs, with the first 1 μs of each replicate considered as equilibration time and removed). From the simulations, mean FRET efficiencies were calculated based on the distance distributions of the fluorescently labelled residues/beads, using the Förster equation. The Förster radius, $R_0$, was set to 6.0 nm as it corresponds to the $R_0$ of the Cy3b-CF660R dye pair used in experiments. All analyses were performed using tools available in the GROMACS suite, custom in-house scripts or MDAnalysis[76].

Contact lifetimes of interactions between the DBD and the AD1, ser-rich region or AD2 domains, for free and bound Sox2, were obtained from calculating the autocorrelation function of contact formation, with a contact between domains defined when the centre of mass of two domains was within 1.0 nm. The interaction lifetimes were obtained by fitting the autocorrelation function using a double

exponential

$$A \exp\left(-\frac{t}{\tau_1}\right) + B \exp\left(-\frac{t}{\tau_2}\right) \qquad (20)$$

## Reporting summary

Further information on research design is available in the Nature Portfolio Reporting Summary linked to this article.

## Data availability

Data supporting the findings of this paper are available from the corresponding authors upon request. The simulation trajectories have been deposited to the Protein Ensemble Database (PED00439 and PED00440). Chemical shift assignments were deposited to the Biological Magnetic Resonance Data Bank (BMRB accession number 51964). Source data are provided with this paper.

## Code availability

Fretica, an add-on package for Mathematica v.12.3 (Wolfram Research) was used for the analysis of single-molecule data (available at https://schuler.bioc.uzh.ch/fretica/).

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

## Acknowledgements

We thank Benjamin Schuler and Daniel Nettels for support using the data analysis toolbox Fretica. We thank Beat Fierz for technical assistance with nucleosome preparation. We thank Magnús Kristjánsson, Erna Magnúsdóttir, and Erik Holmstrom for stimulating discussions, and Matthías Valdimarsson, Sarah Ruidiaz, and Mahtab Hafizi for technical assistance. This work was supported by funding from the European Research Council (ERC StG, 101040601-PIONEER, to P.O.H.), the Icelandic Research Fund (grant nr. 2659105, to P.O.H.), the University of Iceland Doctoral Fund (to S.B.), the Max Planck Society (to J.T.B.), the University of Auckland (to J.A.P.M.), the Novo Nordisk Foundation to the Challenge centre REPIN (#NNF18OC0033926 to B.B.K.), and by

cOpenNMR, an infrastructure granted from the Novo Nordisk Foundation (#NNF18OC0032996).

## Author contributions

S.B., J.A.P.M., A.P., B.B.K., D.M. and P.O.H. designed the study; S.B. and K.S.D. prepared protein and DNA constructs; S.B performed all single-molecule and circular dichroism experiments; S.B. and A.P. performed NMR experiments; J.T.B. provided new analytical tools; J.A.P.M. and D.M. performed simulations; S.B., J.A.P.M., A.P., J.T.B., B.B.K., D.M. and P.O.H. analysed data; S.B. and P.O.H. wrote the manuscript with help from all authors.

## Competing interests

The authors declare no competing interests.
