## [Peer Review File · Nature Communications]

REVIEWER COMMENTS

Reviewer #1 (Remarks to the Author):

In this manuscript, the authors describe biophysical characterizations of intrinsically disordered regions (IDRs) of Sox2 and their interactions with the DNA-binding domain (DBD). Using FRET, NMR, and coarse-grained MD, the authors found that Sox2's IDRs interact with the DBD in the free state but are released upon Sox2's binding to DNA. The work is scientifically sound. However, for the following reasons, I do not recommend this work for publication in Nature Communications.

[1] This work lacks novelty. Prior to this work, many researchers have already reported intra-molecular interactions between IDRs and DBD (or RBD) for many transcription factors and other DNA-binding and RNA-binding proteins (e.g., FoxO3, Hfq, HMGB1, MAX, Nop15, p53, SLBP, SOX-11, U2AF2, and many more). Whenever an IDR competes with DNA (RNA) for a DBD (RBD), the binding of DBD (RBD) to DNA (RNA) should redistribute the structural ensemble of the IDR. Extensive biophysical studies on this effect have already been done for several systems (e.g., p53, HMGB1, FoxO3, U2AF2).

[2] The biological meaning of the intramolecular interactions between the IDR and the DBD of Sox2 is unclear. Although in many other cases, such intra-molecular interactions make the DNA-binding affinity weaker. But in the current case of Sox2, the DNA-binding affinity remains the same even when the IDR is removed. The intramolecular IDR-DBD interactions might impact the affinity for co-activators. However, the current manuscript does not provide any experimental data on the binding affinity for other proteins.

Reviewer #2 (Remarks to the Author):

Bjarnason et al. studied the conformational ensemble of the partially intrinsically disordered pioneer transcription factor Sox2, a factor with important biological functions and applications. The authors pointed out that previous studies revealed that the intrinsically disordered regions (IDRs) of Sox2 do not impact the binding affinity of Sox2 to DNA, which they again confirmed in this study. Though the disordered regions of Sox2 do not contribute to DNA-binding, other biological functions that require interactions of the IDRs with DNA and/or RNA had been found in other studies. The motivation of this study was therefore to characterize the conformational ensemble of Sox2 at the single-residue level to understand whether and how the IDRs of Sox2, which include a short N-terminal stretch (40 aa) and a long C-terminal segment (195 aa), interact with the DNA-binding domain (DBD). This allowed the authors to draw conclusions about the accessibility of the IDRs for other interaction partners. To this end, the authors used an impressive combination of smFRET, NMR, and coarse-grained molecular simulations to obtain a quantitative picture of the Sox2 ensemble in presence and absence of DNA. The authors found that weak contacts between the folded DBD and the C-terminal IDR (C-IDR) are mainly formed by Coulomb-interactions. These interactions are partially screened at physiological ionic strengths. Similarly, also DNA-binding breaks some of the inter-domain contacts that causes a more expanded C-IDR, which then might be accessible for other interaction partners. Using an impressive collection of 11 variants of the proteins, labeled with donor and acceptor dyes, the authors benchmarked a coarse-grained model and revealed the molecular ensemble of Sox2, which is indeed expanded in complex with DNA. Using

chemical shift perturbations (CSPs) and several deletion-constructs, NMR-experiments identified the interaction sites in the C-IDR that contact the DBD and independently confirmed that DBD/C-IDR contacts are mainly formed via electrostatic interactions.

Overall, this is a beautiful study that once more demonstrates the power of integrative approaches in characterizing intrinsically disordered proteins. The experiments are of high quality, the conclusions are justified, and the paper is very well written. I am therefore very positive and recommend publication after the authors clarified a few details.

1. The FRET efficiency histograms sometimes show additional shoulders that remain unaccounted for in the fits of the distributions, (Fig. 1c,d; 2a,b; 3c,d; 4b). I am curious of whether these additional peaks are rather artefacts (What's the origin? – PIE cut?) or whether they might represent additional conformers with longer (> 1 ms) lifetimes that were ignored because of their marginal population.

2. The FRET-lifetime analysis revealed that the C-IDR is an ensemble of conformers. I was wondering whether it would be possible to obtain experimental information on the sampling time (reconfiguration time) of the ensemble, ideally in the presence and absence of DBD and DNA.

3. It would have been nice to mark the DNA-binding site on the DBD in Fig. 2i to judge whether the C-IDR can also contact DBD-regions required for binding. Clearly, removal of the IDR does not change the affinity, yet CSPs might also pick up rather weak transient interactions that would not alter the affinity much.

4. Related to point 3, the authors were surprised about the DNA-induced expansion of the C-IDR. If contacts with the DNA-binding site on the DBD get lost, wouldn't the expansion of the C-IDR be a logical consequence?

5. The CG-simulations contained a DNA model. It would be nice if the authors could also analyze potential interactions of the C-IDR with the DNA. Would it be possible that a breakage of attractive interactions between C-IDR and the DNA-binding site of the DBD is compensated by favorable interactions with the DNA? A reasonable balance between these two interaction types (C-IDR with DBD, C-IDR with DNA) might be another way to explain the lacking impact of the C-IDR on the overall DNA-binding affinity.

6. The above point is interesting as the FRET efficiency of the C-IDR (full-length Sox2) in the DNA-bound state does not only mismatches that of the free C-IDR as the authors explain, but it also does not match that of the C-IDR in Sox2 in the presence of salt. The authors would probably explain the two findings by interactions between AD1 with the DBD that are still present in the DNA-bound state but that are screened at high salt. Yet, an alternative would be direct DNA-interactions of the C-IDR.

7. The short N-terminal tail also seems to interact with the DBD and dissociates upon DNA-binding (Fig. 4b). The authors do not discuss this much, but I was wondering whether the N-terminus has also a function role?

8. The smFRET efficiency histograms show a conformational change in the DBD upon DNA binding (Fig. 4b). Given the known structures of the DBD with and without DNA, is this change to be expected? If so, only one pdb-structure had been used for the simulation with constrained native interactions. Could this structural change potentially impact C-IDR/DBD interactions?

9. I was wondering whether a more quantitative comparison between the CG-simulations and the CSPs from NMR is possible, e.g., correlating the marginalized fraction of contacts for each C-IDR residue with the CSPs?

10. In Fig. 5d, it is unclear what exactly the color scheme reports on. One of the scales range from -0.14 to 0.14 (the other from 0 to 1) and it is unclear whether this is the delta of the fraction of contacts or another quantity.

11. On p. 14 and 16, the lifetime changes of AD1 versus AD2 upon DNA-binding were interpreted as accessibility, which seems incorrect. I'd like to point out that tau from a correlation function is not a lifetime but rather a relaxation time, which is the inverse of the sum of contact formation and dissociation rates. Hence, tau is neither a lifetime (such as fluorescence lifetimes) nor a dwell-time. It is therefore difficult to relate this quantity to accessibility. The fraction of contacts should be a better quantity for judging accessibility.

Reviewer #3 (Remarks to the Author):

In this manuscript, the authors have employed a multi-faceted approach, integrating single-molecule Förster resonance energy transfer (smFRET), nuclear magnetic resonance (NMR), and coarse-grained simulations to probe the conformational landscape of the intrinsically disordered region (IDR) of the transcription factor Sox2 and its alterations upon binding with DNA. The findings illuminate that the intramolecular interactions between the IDR and the structured DNA-binding domain (DBD) of Sox2 do not perturb the DBD's ability to bind DNA. This work provides an in-depth characterization of the intrinsically disordered domain of Sox2, demonstrating the pragmatic utilization of a diverse array of biophysical technologies for IDR research.

Outlined below are specific concerns:

1. The authors have comprehensively characterized the structure and dynamics of the C-IDR and its interaction with DBD. At the same time, they found that C-IDR does not affect the DNA binding of DBD. Therefore, the question arises: what is the functional role of IDR and its interaction with DBD? How strong is the interaction between IDR and DBD? I think the authors should measure the binding affinity between DBD and C-IDR. Is the interaction too weak to induce an effect on DNA binding?

2. The HSQC spectrum of full-length Sox2 shows little dispersion of peaks. On page 6, 2nd line, the author said: "we could assign 275 peaks out of 290...". This means that most of the peaks of the DBD have been assigned, and they also exhibit a narrow distribution. It seems that DBD is unstructured in the full-length protein. In Figure S1, it was stated that "the peaks originating from the DBD overlap well with peaks from an isolated DBD. This indicates that the DBD structure is generally unperturbed by the presence of the C-IDR." However, in Figure 2, the chemical shifts of DBD are obviously affected by the presence of C-IDR. Please address these discrepancies.

3. In the manuscript's text (lines 347-349), the statement, "The difference in contacts between the DBD and C-IDR for free and DNA-bound Sox2 shows that the expansion observed experimentally upon DNA binding is coincident with an increased number of contacts between the N- and C-IDR, and decreased contacts of both IDRs with the DBD (Fig. 5c)." requires clarification. Specifically, the reference to "N- and

C-IDR" necessitates precise definition and alignment with Figure 5c. The figure appears to indicate increased contacts between the DBD and the N-terminal region preceding the AD1 domain, contrasting with the mentioned statement. A coherent explanation reconciling this apparent disparity is warranted.

RESPONSE TO REVIEWERS

We would like to thank all the reviewers for their very helpful and constructive comments on our manuscript. They have prompted us to perform additional experiments and simulations, which are now included in the manuscript's text and figures as additional data and analysis. As a result, we believe the manuscript is substantially more solid and the communication of our results clearer. Below are our detailed responses to all the comments.

REVIEWER COMMENTS

Reviewer #1 (Remarks to the Author):

In this manuscript, the authors describe biophysical characterizations of intrinsically disordered regions (IDRs) of Sox2 and their interactions with the DNA-binding domain (DBD). Using FRET, NMR, and coarse-grained MD, the authors found that Sox2's IDRs interact with the DBD in the free state but are released upon Sox2's binding to DNA. The work is scientifically sound. However, for the following reasons, I do not recommend this work for publication in Nature Communications.

[1] This work lacks novelty. Prior to this work, many researchers have already reported intra-molecular interactions between IDRs and DBD (or RBD) for many transcription factors and other DNA-binding and RNA-binding proteins (e.g., FoxO3, Hfq, HMGB1, MAX, Nop15, p53, SLBP, SOX-11, U2AF2, and many more). Whenever an IDR competes with DNA (RNA) for a DBD (RBD), the binding of DBD (RBD) to DNA (RNA) should redistribute the structural ensemble of the IDR. Extensive biophysical studies on this effect have already been done for several systems (e.g., p53, HMGB1, FoxO3, U2AF2).

We thank the reviewer's comment about the scientific soundness of our work. We considered the reviewer's insight into why our work might lack novelty and thank them for providing an overview of the other studies outlining the effect of D/RBD binding to DNA on the ensembles of the respective IDRs. It was of course not our intention to neglect the wealth of studies that have been done on other systems. Beyond our previous references to recent and relevant work, such as on p53, we have now expanded our discussion to account for the studies performed on the molecular systems that the reviewer mentions.

We are, however, confident that our integrative approach adds both constructively and innovatively to our understanding of transcription factors and to the redistribution of IDRs, in the several ways that we discuss below:

(1) To the best of our knowledge, the redistribution of activation domains in FoxO3 has not been directly addressed and indirect effects upon DNA-binding have been assessed via all-atom MD simulations¹. The simulations, however, due to their high computational cost, could neither reveal a specific redistribution of the protein's transactivation domain nor the effects of different linker compositions on the ensemble (due to high standard deviation). (2) The studies on RNA binding proteins' RRM domains (e.g., U2AF2²) within the context of intrinsic disorder focus mainly on the relative positioning of the folded domains (indirectly probed by changes in K_D). In those studies, disordered linkers are treated as a "bystander" (rather than an active element for molecular recognition, as in the case of Sox2). (3) In several other cases, focus has been on the conformational ensembles by means of NMR and MD simulations, but they never quantitatively

couple experiments and simulations^{3,4}, as we do through the direct crosstalk of smFRET and computational efforts. (4) Other studies that we were able to find in the literature are too low-resolution to specifically focus on conformational ensembles^{5,6}.

Importantly, we believe that our study highlights the advanced capabilities of reconstructing experimentally-driven ensembles of full-length IDPs: pinpointing not only the redistribution of activation domains (AD) but also their relative distance and positioning (in this case across the full length of Sox2). To the best of our knowledge, the strongest evidence of AD redistribution has been provided for p53 by Krois et al.⁷ (in a truly beautiful study), via extensive selective isotope labelling of the full-length tetrameric p53. However, we note that there are several differences between that study and ours. The activation domains in the ~60 residue NTAD of p53 are very close to each other (only 10 residues separate the two) and NMR (via PRE experiments) would hint towards a redistribution of the AD1 and AD2 domains within a somewhat shorter distance range (up to ~20Å) due to the different resolution range of PRE experiments (as compared to smFRET which can detect FRET changes for up to ~100Å). More relevantly, to the best of our knowledge, previous studies of IDRs responding to DNA binding seem to be accompanied by substantial changes in binding constants, which we do not observe for Sox2. All those proteins are characterized by a large content of charged residues in the relevant IDRs, or low complexity polyglycine/polyserine linkers (acting like spacers between folded domains). Sox2 on the other hand, represents an example of few and well distributed charges. What many of the other, previously studied, proteins have in common (e.g., Nop15⁸, SLBP⁴, Sox11⁵) are short and highly charged patches in their IDRs: these interact strongly with the multitude of positive charges in the respective DBDs and influence nucleic acid binding. In the case of p53 the electrostatic nature of the disordered NTD p53 domain is directly called into play by the authors (Krois et al.⁷) to explain the observed interplay between NTD and DBD, with the NTD containing several aspartates in the AD1 and AD2. **Conversely, our study provides evidence that even though Sox2 has very few and well distributed charges in its C-IDR we observe a strong effect on overall dimensions (see Fig. 2f) without effects on DNA binding.** It is likely that the distribution of charges plays a role in how dimensions are modulated upon DNA binding, and phosphorylation would be further expected to affect the relative accessibility changes between the two activation domains: an avenue we are currently pursuing. Therefore, we argue that the case we present here for Sox2 is novel, as outlined in the manuscript.

We have included a reference to these studies in the introduction and have added a sentence to the discussion on p. 16 to solidify one of our main conclusions: “This is a noteworthy observation: even though the charges in the C-IDR are relatively sparse and well distributed, their interactions with the DBD still confer a strong effect on the overall dimension of the protein.”

Overall, we believe that an important additional element of the novelty of our work involves the capacity to grasp and show conformational changes in Sox2 upon DNA binding, as revealed via the tight combination of experiments and detailed dynamic structural modelling. Given the distance between DBD and activation domains, and the relatively few charges of the Sox2 C-IDR, the observation that DNA binding leads to differential accessibility of Sox2 activation domains is non-trivial, and to our knowledge has not been demonstrated previously. The findings we present can thus provide an important basis to test the role of disorder within supramolecular assemblies (such as nucleosomes and chromatin) relevant to transcription, with new hypotheses to further scrutinize the Sox2 “molecular ecosystem”. We are currently working on both understanding better the role of charges on dimensions and binding partner selection.

[2] The biological meaning of the intramolecular interactions between the IDR and the DBD of Sox2 is unclear. Although in many other cases, such intra-molecular interactions make the DNA-binding affinity weaker. But in the current case of Sox2, the DNA-binding affinity remains the same even when the IDR is removed. The intramolecular IDR-DBD interactions might impact the affinity for co-activators. However, the current manuscript does not provide any experimental data on the binding affinity for other proteins.

As the reviewer correctly points out, the biological meaning of our results is currently unknown. However, a large contribution to deciphering biological meaning comes from a detailed biophysical characterization of the interactions of Sox2 and other relevant cellular partners (i.e., DNA). Within this preamble, it is intriguing that the intramolecular interactions observed for Sox2 do not affect DNA binding, attesting also to the novelty of the findings. As we speculate in the paper, these types of electrostatic interactions offer the possibility to fine-tune structure and accessibility of IDRs, e.g., through post-translational modifications. This can have effects both on the lifetime of the protein, i.e., the degradation rate, and the interaction pattern with binding partners and the transcriptional machinery. It may also serve to both increase solubility of the C-IDR (it is highly insoluble on its own) and reduce interactions with binding partners before the right promoter is stably bound. Our structural model creates the important fundamentals to better understand the function of Sox2 within increasingly complex molecular contexts, e.g., featuring other binding partners. Even though we are already invested in answering these questions, dissecting these finer details of molecular interactions between IDRs and binding partners is an extensive and challenging task, and beyond the scope of this paper.

Reviewer #2 (Remarks to the Author):

Bjarnason et al. studied the conformational ensemble of the partially intrinsically disordered pioneer transcription factor Sox2, a factor with important biological functions and applications. The authors pointed out that previous studies revealed that the intrinsically disordered regions (IDRs) of Sox2 do not impact the binding affinity of Sox2 to DNA, which they again confirmed in this study. Though the disordered regions of Sox2 do not contribute to DNA-binding, other biological functions that require interactions of the IDRs with DNA and/or RNA had been found in other studies. The motivation of this study was therefore to characterize the conformational ensemble of Sox2 at the single-residue level to understand whether and how the IDRs of Sox2, which include a short N-terminal stretch (40 aa) and a long C-terminal segment (195 aa), interact with the DNA-binding domain (DBD). This allowed the authors to draw conclusions about the accessibility of the IDRs for other interaction partners. To this end, the authors used an impressive combination of smFRET, NMR, and coarse-grained molecular simulations to obtain a quantitative picture of the Sox2 ensemble in presence and absence of DNA. The authors found that weak contacts between the folded DBD and the C-terminal IDR (C-IDR) are mainly formed by Coulomb-interactions. These interactions are partially screened at physiological ionic strengths. Similarly, also DNA-binding breaks some of the inter-domain contacts that causes a more expanded C-IDR, which then might be accessible for other interaction partners. Using an impressive collection of 11 variants of the proteins, labeled with donor and acceptor dyes, the authors benchmarked a coarse-grained model and revealed the molecular ensemble of Sox2, which is indeed expanded in complex with DNA. Using chemical shift perturbations (CSPs) and several deletion-constructs, NMR-experiments identified the interaction sites in the C-IDR that contact the DBD and

independently confirmed that DBD/C-IDR contacts are mainly formed via electrostatic interactions. Overall, this is a beautiful study that once more demonstrates the power of integrative approaches in characterizing intrinsically disordered proteins. The experiments are of high quality, the conclusions are justified, and the paper is very well written. I am therefore very positive and recommend publication after the authors clarified a few details.

We would like to thank the reviewer for the positive assessment of our work.

1. The FRET efficiency histograms sometimes show additional shoulders that remain unaccounted for in the fits of the distributions, (Fig. 1c,d; 2a,b; 3c,d; 4b). I am curious of whether these additional peaks are rather artefacts (What's the origin? – PIE cut?) or whether they might represent additional conformers with longer (> 1 ms) lifetimes that were ignored because of their marginal population.

We thank the reviewer for pointing this out. We have now added a note to the caption of Figure 1 referencing a new supplementary figure (Fig. S1) that shows that these peaks are indeed artefacts originating from the way we filter out bursts from donor-only molecules using pulsed interleaved excitation (PIE). Fluorophore labelling of large proteins such as Sox2 is tricky and separating dual-labelled proteins from donor-only-labelled proteins can be challenging. Therefore, donor-only bursts can persist even after PIE-filtering. It should be noted that residual bursts from donor-only molecules do not affect the analysis of FRET efficiencies of double-labelled samples.

2. The FRET-lifetime analysis revealed that the C-IDR is an ensemble of conformers. I was wondering whether it would be possible to obtain experimental information on the sampling time (reconfiguration time) of the ensemble, ideally in the presence and absence of DBD and DNA.

We thank the reviewer for suggesting this additional experiment. We have now performed nanosecond fluorescence correlation spectroscopy (nsFCS) of Sox2 fluorescently labelled in the C-IDR, both on the full-length construct in presence and absence of DNA, and on the isolated C-IDR. We then analyzed the donor-acceptor cross-correlation functions which allowed us to determine the reconfiguration time⁹. This analysis yielded reconfiguration times on a ~100 ns timescale, in agreement with the fluorescence lifetime data (Fig. S9). Interestingly, the reconfiguration time is somewhat shorter for the isolated C-IDR compared to free full-length Sox2, presumably due to the lack of intramolecular interactions with the DBD.

These data have now been added to a new supplementary Figure S6 and reported in the results section: *“To quantify the dynamics, we performed nanosecond fluorescence correlation spectroscopy (nsFCS) experiments of Sox2 in absence and presence of DNA, probing the C-IDR dynamics (Fig. S6). Fitting the anti-correlated donor-acceptor cross-correlation functions, which decay on the timescale of interdye distance fluctuations, allowed us to determine the reconfiguration time (τ_r) of the C-IDR (Methods). In agreement with the fluorescence lifetime analysis, τ_r is similar in the absence and presence of DNA (172 ns and 184 ns, respectively) whereas the isolated C-IDR reconfigures slightly faster ($\tau_r \sim 105$ ns), presumably due to the lack of the neighbouring DBD to interact with.”* We have also added a description of the methodology in the Methods section.

3. It would have been nice to mark the DNA-binding site on the DBD in Fig. 2i to judge whether the C-IDR can also contact DBD-regions required for binding. Clearly, removal of the IDR does not change the affinity, yet CSPs might also pick up rather weak transient interactions that would not alter the affinity much.

The locations of the DNA-binding residues have now been added to Fig. 2i. The CSPs show that contacts are indeed also lost in this region but are presumably weak and transient, as pointed out by the reviewer.

4. Related to point 3, the authors were surprised about the DNA-induced expansion of the C-IDR. If contacts with the DNA-binding site on the DBD get lost, wouldn't the expansion of the C-IDR be a logical consequence?

We agree with the reviewer that "surprising" may be an overstatement. We were initially surprised by the degree of expansion, which corresponds to 25-30% in terms of end-to-end distance but agree that an expansion would be expected due to loss of contacts. We have revised this sentence accordingly: *Given the lack of effects on binding affinity, We were surprised to observe a substantial change in FRET efficiency; the C-IDR expanded considerably upon binding DNA, with FRET decreasing from 0.43 to 0.28 (Fig. 3c).*"

5. The CG-simulations contained a DNA model. It would be nice if the authors could also analyze potential interactions of the C-IDR with the DNA. Would it be possible that a breakage of attractive interactions between C-IDR and the DNA-binding site of the DBD is compensated by favorable interactions with the DNA? A reasonable balance between these two interaction types (C-IDR with DBD, C-IDR with DNA) might be another way to explain the lacking impact of the C-IDR on the overall DNA-binding affinity.

We thank the reviewer for the excellent points raised and we ran additional contact analyses to assess and compare the interaction of the C-IDR with either the DBD in the unbound ensemble or the DNA in the bound ensemble. Interestingly, as hinted in the comment, the distribution of the DNA/C-IDR fraction of contacts shifts to lower values but with a tail towards higher fraction of contacts made between the DNA and the N-terminal end of the C-IDR. More interestingly, this region features a stretch of conserved positively charged residues that tends to interact stably with the DNA in our simulations.

Due to the simple CG potential used in the simulations, a direct quantitative comparison of the energetics of these changes with the experimentally measured K_D is non-trivial. However, it is intriguing to note that these interactions, being of strong electrostatic nature, are specific and resilient and they could enthalpically counteract the less unspecific and more transient interactions of the rest of the C-IDR with the DBD. It is noteworthy that region 110-140, albeit disordered, contains conserved positively charged residues with a RPRRK motif that, in all the resolved structures of Sox2 bound to DNA (in several contexts, including nucleosomal¹⁰), strongly associates with DNA with geometrically directed electrostatic interactions that standardize the modelled conformations of this short intrinsically disordered stretch compared to the DNA double helix.

We have now added the boxplots of the contact distributions into a new supplementary Figure S11 and cite it in the manuscript as: *"We also analyzed whether loss of C-IDR contacts with the DBD might be*

accompanied by formation of new contacts with the DNA but there was no enrichment in contact formation beyond a short ~20-residue stretch immediately flanking the DBD which is known to stably bind into the DNA major groove^{28,31}(Fig. S11).”.

6. The above point is interesting as the FRET efficiency of the C-IDR (full-length Sox2) in the DNA-bound state does not only mismatches that of the free C-IDR as the authors explain, but it also does not match that of the C-IDR in Sox2 in the presence of salt. The authors would probably explain the two findings by interactions between AD1 with the DBD that are still present in the DNA-bound state but that are screened at high salt. Yet, an alternative would be direct DNA-interactions of the C-IDR.

Prompted by this remark, we did further analysis on our simulations at different salt concentrations and looked specifically at the contacts of AD1 and AD2 in the free and DNA-bound states. The probability density distributions of the number of contacts as a function of the salt concentration indicates that long range electrostatic screening reduces the interaction of DBD and the C-IDR.

We have now added the afore-mentioned probability density distributions to the manuscript as new panels in supplementary figure S10 and have modified the text by adding the following sentence: *“Interestingly, an increase in salt concentration screens the interaction between the DBD and the AD1/AD2 domains with an effect that is proximity-dependent and more pronounced for AD1 (Fig. S10c)”.*

Nonetheless, this effect is likely linked to the observed more specific electrostatic interactions that the N-terminal end of the C-IDR makes with the DNA, and the effect on the K_D is likely dependent on both, to different extents.

7. The short N-terminal tail also seems to interact with the DBD and dissociates upon DNA-binding (Fig. 4b). The authors do not discuss this much, but I was wondering whether the N-terminus has also a function role?

This is indeed an interesting observation, which prompted us to look deeper into the literature. Unfortunately, the literature does not provide much information on the function of the N-IDR. There is some evidence that the N-IDR assists in the interaction with the transcription factor Sall4¹¹. It is entirely possible that these interactions would also be modulated upon DNA binding but given the lack of strong evidence for the N-IDR function and that it is much shorter than the C-IDR (40 vs 200 amino acids), we have decided not to discuss this point further, to not be too speculative at this stage. It will be interesting to investigate this in future studies. We have also added a reference to the above study and added a sentence to the introduction to read *“Little is known about the function of the short N-IDR but there is evidence that it is important for interactions with other transcription factors^{16”.}*

8. The smFRET efficiency histograms show a conformational change in the DBD upon DNA binding (Fig. 4b). Given the known structures of the DBD with and without DNA, is this change to be expected? If so, only one pdb-structure had been used for the simulation with constrained native interactions. Could this structural change potentially impact C-IDR/DBD interactions?

It is true that we used the same DBD structure for simulating both free and DNA bound Sox2; a structure determined by cryo electron microscopy (PDB 6T7B). We proceeded to do that because *a)* the only structure

of free Sox2 (PDB 2LE4) is an NMR structure that was deposited in the PDB in 2011 but has so far not been published and therefore we cannot adequately assess its quality, and *b*) because the two structures are highly similar. Both structures (free and DNA-bound) include only residues 39-115 but our FRET dyes are placed at residues 37 and 120, covering more disordered segments and therefore the slight conformational change that we observe in our experiments upon DNA binding may well come from regions outside the structured regions. Given the considerable FRET change in the long C-IDR, the slight FRET shift within the DBD would not be expected to have much effect on what we observe with C-IDR/DBD interactions.

Nevertheless, to specifically address whether the DBD structure affects the outcome of the simulations, we performed additional simulations, this time using the unpublished NMR structure of free Sox2 (PDB 2LE4). The FRET efficiencies obtained from this set of simulations yielded a nearly identical match with the experiments when compared to the set of simulations where the structure of the Sox2 DBD bound to nucleosomal DNA (PDB 6T7B) was used. Both sets of simulations are thus in agreement with the experiments, with the concordance correlation coefficients approximately 3% apart. We noted that we had previously incorrectly reported the Pearson correlation coefficient in the paper but now rightly report the concordance correlation coefficient for all simulations.

Given the minimal differences between the two simulations and due to the NMR structure still being unpublished, we think it is warranted to keep our current comparison using the same structure for both free and DNA-bound Sox2. We have put the simulation agreement for structure 2LE4 into supplementary Fig. S10, along with an alignment of the two DBD structures, and we have added the following sentence in the results on p.13 *"Since no published structures are available for free Sox2, we used the same structure for both free and DNA-bound Sox2³¹. However, a simulation using a thus far unpublished NMR structure of free Sox2 DBD deposited in the PDB (PDB code 2LE4) yielded near identical results (Fig. S10)."*

9. I was wondering whether a more quantitative comparison between the CG-simulations and the CSPs from NMR is possible, e.g., correlating the marginalized fraction of contacts for each C-IDR residue with the CSPs?

We would like to thank the reviewer for this excellent suggestion, which makes the comparison between experiments and simulations clearer. To account for the different environment due to the DNA, we calculated the difference in contacts between DBD and C-IDR and added the difference in contacts between C-IDR and DNA. We now have a new Fig. S11 showing how the CSPs from the NMR experiments compare with the difference in fraction of contacts between free and DNA-bound Sox2 from the simulations, which shows good agreement between the two approaches. Given the different nature of chemical shifts calculated by NMR compared to what low-resolution CG simulations can yield, we do not expect a quantitative match between the two. Nevertheless, we note that the qualitative match between the CSPs and the contacts sampled by simulations is reassuring.

10. In Fig. 5d, it is unclear what exactly the color scheme reports on. One of the scales range from -0.14 to 0.14 (the other from 0 to 1) and it is unclear whether this is the delta of the fraction of contacts or another quantity.

We agree that the scale is unfortunate and confusing. We have now made this figure clearer by including only the most relevant sausage plot (previously panel e, now panel d), which is the one that depicts most

clearly the results from panel 5c. To alleviate any further confusion, we have changed the coloring of this plot such that it now reflects the colors in the contact map (panel 5c).

In addition, and prompted by the next comment by the reviewer, we have replaced panels 5f,g with another plot that shows more clearly the different fraction of contacts made by AD1 and AD2 (see our response to comment #11 below).

11. On p. 14 and 16, the lifetime changes of AD1 versus AD2 upon DNA-binding were interpreted as accessibility, which seems incorrect. I'd like to point out that tau from a correlation function is not a lifetime but rather a relaxation time, which is the inverse of the sum of contact formation and dissociation rates. Hence, tau is neither a lifetime (such as fluorescence lifetimes) nor a dwell-time. It is therefore difficult to relate this quantity to accessibility. The fraction of contacts should be a better quantity for judging accessibility.

We thank the reviewer for pointing this out and agree that what we have plotted is a relaxation time. We apologize for the confusion this might have created. We have now also plotted the difference in fraction of contacts for AD1 and AD2. Prompted by the reviewer's comment we have decided to replace panel e) of Figure 5, which now shows violin plots of contact distributions as box plots within the probability densities of the violin plots, and we have moved the relaxation time plots to the supplementary material. The new violin plots compute the number of contacts of AD1 and AD2 with the DNA, showing the relative change being larger for AD2 overall, as also shown in the contact map (Figure 5c). We hope that this now resolves the confusion giving the reader a clearer picture of how the relative accessibility might change for both domains.

Reviewer #3 (Remarks to the Author):

In this manuscript, the authors have employed a multi-faceted approach, integrating single-molecule Förster resonance energy transfer (smFRET), nuclear magnetic resonance (NMR), and coarse-grained simulations to probe the conformational landscape of the intrinsically disordered region (IDR) of the transcription factor Sox2 and its alterations upon binding with DNA. The findings illuminate that the intramolecular interactions between the IDR and the structured DNA-binding domain (DBD) of Sox2 do not perturb the DBD's ability to bind DNA. This work provides an in-depth characterization of the intrinsically disordered domain of Sox2, demonstrating the pragmatic utilization of a diverse array of biophysical technologies for IDR research.

We would like to thank the reviewer for the positive comments on our manuscript.

Outlined below are specific concerns:

1. The authors have comprehensively characterized the structure and dynamics of the C-IDR and its interaction with DBD. At the same time, they found that C-IDR does not affect the DNA binding of DBD. Therefore, the question arises: what is the functional role of IDR and its interaction with DBD? How strong

is the interaction between IDR and DBD? I think the authors should measure the binding affinity between DBD and C-IDR. Is the interaction too weak to induce an effect on DNA binding?

The reviewer raises very good questions. The function of the IDRs of transcription factors has traditionally been linked to activation or repression of transcription through binding of protein partners. For Sox2, this has been well established in terms of transcriptional activation, especially for genes important for pluripotency. However, it is also well established that IDRs of transcription factors can be multifunctional (see for example a recent review we wrote on the subject¹²) and this may indeed be the case for Sox2 as well. Maintaining intramolecular interactions with the DBD may be advantageous to counteract premature degradation, modulate binding to partners, or for maintaining solubility of an otherwise insoluble C-terminal. We have tried our best to mention these and other points explicitly in the Discussion on p.16.

The lack of effects on DNA binding affinity indeed seems to indicate that the interaction is weak. As suggested by the reviewer, we decided to measure the binding affinity between the C-IDR and the DBD. We therefore performed additional NMR experiments and recorded ¹⁵N-HSQC of ¹⁵N-labelled C-IDR with increasing concentrations of unlabelled N-DBD. This allowed us to follow the shift changes of highly perturbed residues as a function of DBD concentration and to estimate the dissociation constant, K_D . As expected, fits of the resulting binding isotherms revealed weak binding with a K_D of $\sim 80 \mu\text{M}$, almost 5 orders of magnitude weaker than the binding affinity of the DBD for DNA, and supporting our conclusion that interdomain interactions do not perturb DNA binding. We have added this new data as two additional panels in supplementary Figure S4 and we also added the following sentence to the results section: “*Using the chemical shift changes of highly perturbed residues, we could also estimate the dissociation constant, K_D , for the complex in trans to be $80 \pm 4 \mu\text{M}$ (Fig. S4).*”. The NMR experiments were performed with a lower salt concentration (high salt concentrations lowers the sensitivity of cryoprobes due to increased resistance) than those assumed under physiological conditions and the actual binding affinity may thus be somewhat lower due to increased charge screening. We surmise, however, that the tethering of the two domains results in high effective concentrations counteracting this, but it is likely that the interactions would nonetheless be transient and weak.

2. The HSQC spectrum of full-length Sox2 shows little dispersion of peaks. On page 6, 2nd line, the author said: “we could assign 275 peaks out of 290...”. This means that most of the peaks of the DBD have been assigned, and they also exhibit a narrow distribution. It seems that DBD is unstructured in the full-length protein. In Figure S1, it was stated that “the peaks originating from the DBD overlap well with peaks from an isolated DBD. This indicates that the DBD structure is generally unperturbed by the presence of the C-IDR.” However, in Figure 2, the chemical shifts of DBD are obviously affected by the presence of C-IDR. Please address these discrepancies.

We apologize for not explaining this well enough in the text and we have now taken steps to make this point clearer. The HSQC shown in Figure 1 of the full-length protein is shown at such a contour level that only the strong peaks originating from the C-IDR are visible, and they have a particularly narrow distribution as expected. If we increase the contour level, – or the temperature– we can indeed see most of the peaks for the DBD which are well dispersed, indicating a well-folded domain, and they overlap sufficiently well with the HSQC of the isolated DBD to allow for transfer of assignments (see text on p.6, l.148, and Figure S2 in revised manuscript). The peak intensities for the DBD within the full-length protein are deliberately not seen at the low temperature where we assign the IDR, needed to slow down amide hydrogen exchange. This is

especially evident in the 3D-spectra where all peaks for the DBD (within the full-length protein) are not observable. If we compare the chemical shifts of the DBD within full-length Sox2 and the isolated DBD, they are affected to the degree shown in Figure 2 but still not to the degree that it precluded transfer of assignments from the spectra of the isolated DBD. It should be noted that the CSPs of the DBD when titrated with the C-IDR are small, in line with the weak interaction, but specific (e.g., note the absence of CSPs in the N-IDR). We have now added to the new Figure S2 additional panels to illustrate specific examples of chemical shift perturbations.

3. In the manuscript's text (lines 347-349), the statement, "The difference in contacts between the DBD and C-IDR for free and DNA-bound Sox2 shows that the expansion observed experimentally upon DNA binding is coincident with an increased number of contacts between the N- and C-IDR, and decreased contacts of both IDRs with the DBD (Fig. 5c)." requires clarification. Specifically, the reference to "N- and C-IDR" necessitates precise definition and alignment with Figure 5c. The figure appears to indicate increased contacts between the DBD and the N-terminal region preceding the AD1 domain, contrasting with the mentioned statement. A coherent explanation reconciling this apparent disparity is warranted.

We thank the reviewer for pointing this out and fully agree that the statement is confusing. What we are referring to here is the N-terminal IDR (N-IDR, residues 1-40) and the C-terminal IDR (C-IDR, residues 120-317). It is true that overall contacts between both the N-IDR and C-IDR with the DBD are reduced in the DNA-bound state, which coincides with a slight increase in contacts between the two IDRs. We note that there is a small region of the C-IDR, namely residues 130-160 that lead to and are part of AD1, that have increased contacts with the DBD. To clarify these interactions, we have changed the color scheme in panel 5d to reflect the colors in the contact map (panel 5c).

We note that the interactions of this region are constrained by stable interactions between the region of 110-125 (which is rich in positively charged residues) and the DNA. As per comment #5 in response to reviewer #2, we see this region to interact stably with the DNA, as observed by the bright spot that can be seen in the contact map shown in Fig. 5c. Positively charged motifs in the region immediately downstream the DBD, are interestingly conserved across Sox proteins (and in Sox2 from different species), albeit being disordered.

We have now addressed this point more clearly in the text by replacing panel 5f with violin plots reporting the redistribution of contacts for both activation domains upon DNA binding. The violin plots show how the relative redistribution affects AD2 more greatly than AD1. Overall, the violin plots show that AD1 increases its accessibility overall, despite some additional contacts formed at the N-terminal end (as shown by the coloring in our sausage plot in panel 5d), due to the constraints triggered by the strong and stable interaction between the DNA and the positively charged conserved residues upstream the C-IDR (110-125).

We have also addressed this issue by defining the IDRs more explicitly in the introduction with the following revision: "*Sox2 has 317 residues and consists of a small HMG-box DBD¹⁵ flanked N-terminally by a short 40-residue low-complexity stretch and C-terminally by a long ~200-residue region, both of which are predicted to be disordered (N-IDR and C-IDR, respectively)(Fig. 1a).*" and revising the sentence in the corresponding results section: "*The difference in contacts between the DBD and C-IDR for free and DNA-bound Sox2 shows that the expansion observed experimentally upon DNA binding is coincident with an increased number of contacts between the N-IDR and C-IDR, and decreased overall contacts of both IDRs with the DBD (Fig. 5c).*".

We also labelled Fig. 5c more clearly to show where the N-IDR, DBD, and C-IDR are defined. Finally, we expanded the discussion to include a note on the conserved positive stretch C-terminal to the DBD that interacts with the DNA: “For example, there is a conserved positive region flanking the DBD on the C-terminal side that we observe to stably interact with the DNA in our simulations, in agreement with previous studies²⁸. Phosphorylations in this region (e.g., on Thr116⁵⁷) would be expected to decrease interactions with DNA, potentially increasing the accessibility of AD1, while a phosphorylation further downstream in the sequence (e.g., Ser251) might increase interactions with the DBD leading to decreased accessibility of AD2. Increased interactions between the C-IDR and DBD might in some cases lead to less efficient DNA binding, which could explain why Sox2 binds certain enhancers less when phosphorylated in Ser251 which is close to AD2⁵⁷.”

We hope that this apparent contradiction is now resolved for the reviewer and that the modifications to the text have made this point clearer for the reader.

REFERENCES

- 1 Weinzierl, R. O. J. Molecular Dynamics Simulations of Human FOXO3 Reveal Intrinsically Disordered Regions Spread Spatially by Intramolecular Electrostatic Repulsion. *Biomolecules* **11**, 856 (2021).
- 2 Kang, H. S. *et al.* An autoinhibitory intramolecular interaction proof-reads RNA recognition by the essential splicing factor U2AF2. *Proc Natl Acad Sci U S A* **117**, 7140-7149 (2020).
- 3 Wang, X. *et al.* Dynamic Autoinhibition of the HMGB1 Protein via Electrostatic Fuzzy Interactions of Intrinsically Disordered Regions. *J Mol Biol* **433**, 167122 (2021).
- 4 Zaharias, S., Fargason, T., Greer, R., Song, Y. & Zhang, J. Electronegative clusters modulate folding status and RNA binding of unstructured RNA-binding proteins. *Protein Sci* **32**, e4643 (2023).
- 5 Wiebe, M. S., Nowling, T. K. & Rizzino, A. Identification of novel domains within Sox-2 and Sox-11 involved in autoinhibition of DNA binding and partnership specificity. *J Biol Chem* **278**, 17901-17911 (2003).
- 6 Kawase, T., Sato, K., Ueda, T. & Yoshida, M. Distinct domains in HMGB1 are involved in specific intramolecular and nucleosomal interactions. *Biochemistry* **47**, 13991-13996 (2008).
- 7 Krois, A. S., Dyson, H. J. & Wright, P. E. Long-range regulation of p53 DNA binding by its intrinsically disordered N-terminal transactivation domain. *Proc Natl Acad Sci U S A* **115**, E11302-E11310 (2018).
- 8 Zaharias, S. *et al.* Intrinsically disordered electronegative clusters improve stability and binding specificity of RNA-binding proteins. *J Biol Chem* **297**, 100945 (2021).
- 9 Gopich, I. V., Nettels, D., Schuler, B. & Szabo, A. Protein dynamics from single-molecule fluorescence intensity correlation functions. *J Chem Phys* **131**, 1-5 (2009).
- 10 Dodonova, S. O., Zhu, F., Dienemann, C., Taipale, J. & Cramer, P. Nucleosome-bound SOX2 and SOX11 structures elucidate pioneer factor function. *Nature* **580**, 669-672 (2020).
- 11 Cox, J. L., Mallanna, S. K., Luo, X. & Rizzino, A. Sox2 uses multiple domains to associate with proteins present in Sox2-protein complexes. *PLoS One* **5**, e15486 (2010).
- 12 Mar, M., Nitsenko, K. & Heidarsson, P. O. Multifunctional Intrinsically Disordered Regions in Transcription Factors. *Chemistry* **29**, e202203369 (2023).

REVIEWERS' COMMENTS

Reviewer #3 (Remarks to the Author):

The authors have addressed all my concerns in the revised manuscript.

Editorial note: Reviewer #2 provided confidential comments visible only to the editors but considers their points addressed and recommends publication of this manuscript.